# Predicting the antigenic evolution of SARS-COV-2 with deep learning

Wenkai Han[1,2,9], Ningning Chen [1,2,9], Xinzhou Xu [3,4,9], Adil Sahil[1,2], Juexiao Zhou[1,2], Zhongxiao Li [1,2], Huawen Zhong[2], Elva Gao[5], Ruochi Zhang[6], Yu Wang[6], Shiwei Sun[7,8] ✉, Peter Pak-Hang Cheung [3,4] ✉ & Xin Gao [1,2] ✉

The relentless evolution of SARS-CoV-2 poses a significant threat to public health, as it adapts to immune pressure from vaccines and natural infections. Gaining insights into potential antigenic changes is critical but challenging due to the vast sequence space. Here, we introduce the Machine Learning-guided Antigenic Evolution Prediction (MLAEP), which combines structure modeling, multi-task learning, and genetic algorithms to predict the viral fitness landscape and explore antigenic evolution via in silico directed evolution. By analyzing existing SARS-CoV-2 variants, MLAEP accurately infers variant order along antigenic evolutionary trajectories, correlating with corresponding sampling time. Our approach identified novel mutations in immunocompromised COVID-19 patients and emerging variants like XBB1.5. Additionally, MLAEP predictions were validated through in vitro neutralizing antibody binding assays, demonstrating that the predicted variants exhibited enhanced immune evasion. By profiling existing variants and predicting potential antigenic changes, MLAEP aids in vaccine development and enhances preparedness against future SARS-CoV-2 variants.

As the number of infection cases increased and the SARS-COV-2 spread globally, novel mutations in the virus genome emerged[1–4]. At the time of April 2022, there are more than one million variants in the virus genome identified and uploaded to the Global Initiative on Sharing Avian Influenza Database (GISAID). The mutations often implicate the changes to the SARS-COV-2 properties[3]. Although most mutations decrease the virulence and transmissibility of the virus[5], some individual or combinatorial mutations substantially improve the transmissibility with enhanced cell entry efficacy[6], or ablate the neutralizing antibodies response elicited by infection or vaccine[1,7], resulting in high-risk variants. For example, the Alpha (B.1.1.7) variant of concern (VOC) spread worldwide through a higher human ACE2-binding affinity and transmissibility than the original Wuhan strain[8]. The Beta and the Gamma lineage abolished the neutralizing antibodies elicited by approved COVID-19 vaccines[9]. The Delta variant became a dominant strain worldwide with the increased transmissibility and morality[10,11]. Recently, the heavily mutated Omicron variant caused new waves due to the extremely high rate of spread and the ability to evade the double-vaccinated person[12].

A substantial fraction of neutralizing antibodies, including monoclonal antibodies and those induced by the vaccines, target the spike receptor-binding domain (RBD)[13–15]. Antibodies targeting the

[1]Computer Science Program, Computer, Electrical and Mathematical Sciences and Engineering Division, King Abdullah University of Science and Technology (KAUST), Thuwal 23955-6900, Kingdom of Saudi Arabia. [2]Computational Bioscience Research Center, King Abdullah University of Science and Technology (KAUST), Thuwal 23955-6900, Kingdom of Saudi Arabia. [3]Department of Chemical Pathology, Faculty of Medicine, Chinese University of Hong Kong, Hong Kong, China. [4]Li Ka Shing Institute of Health Sciences, Chinese University of Hong Kong, Hong Kong, China. [5]The KAUST School, King Abdullah University of Science and Technology (KAUST), Thuwal 23955-6900, Kingdom of Saudi Arabia. [6]Syneron Technology, Guangzhou 510000, China. [7]Key Lab of Intelligent Information Processing, Institute of Computing Technology, Chinese Academy of Sciences, Beijing 100190, China. [8]University of Chinese Academy of Sciences, Beijing 100049, China. [9]These authors contributed equally: Wenkai Han, Ningning Chen, Xinzhou Xu. ✉e-mail: dwsun@ict.ac.cn; ppcheung@cuhk.edu.hk; xin.gao@kaust.edu.sa

RBD have been divided into four categories according to their binding epitopes[16]. Class 1 and class 2 antibodies bind the surface of the receptor-binding motif (RBM) and thus compete with ACE2 for RBD binding. Mutations in the RBM region, in turn, decreased neutralization by these antibodies. Class 3 antibodies bind the opposite side of the RBM, contain less overlap with the ACE2-binding footprint, provide the potential for synergistic effects when combined with Class 1 and 2 antibodies for intercepting ACE2 binding[17]. Class 4 antibodies target a highly conserved region among sarbecoviruses and thus are generally more resistant to the variants[18]. However, emerging viral lineages such as Omicron and BA.2 can still lead to a substantial loss of neutralization[19].

Understanding the role of the mutations and how they are linked to transmissibility and immune escape are thus of great importance. There have been an expanding set of analyses characterizing these problems[5,18,20–23]. Starr et al.[5] and Greaney et al.[18] performed deep mutational scanning (DMS) on the entire Spike RBD sequences of SARS-COV-2 on the yeast surface to determine the impact of single-position substitutions on the binding ability to ACE2 and monoclonal antibodies. These assayed experiments provide a unique resource for understanding the properties of variants. However, the wet-lab experiments are resource and time-consuming, and cannot be scaled to the large protein sequence space. Maher et al.[23] characterized the potential risks of the single-position substitutions with a computational model and forecasted the driver mutations that may appear in emerging VOCs. Despite their effectiveness in modeling the risks at the single-mutant level, the newly emerging VOCs (e.g., Delta, Omicron) often possess multiple mutations in the RBD region, which directly influences the ACE2 binding and antibody escape. For example, the Omicron variant contains 15 mutations in the RBD region and obtains considerable antigenic escape ability[24]. Moreover, the effects of mutations are context-dependent, such that the epistatic interactions among the mutations limit the application scenario of the single-mutant-based methods[25].

The sequence space of protein variants grows exponentially when multiple mutations are considered, while measuring the functionality of the variant sequences far exceeding the capacity of wet-lab experiments. Machine learning methods have been proposed for solving the problem[26–28]. Alexander et al.[29] trained a large-scale transformer model with the self-supervised protein language modeling objective, while the model can infer the effects of mutations without supervision. Chloe et al.[30] combined linear regression with the Potts model, resulting in a data-efficient variant fitness inference model. These models have been proven to be effective in the protein engineering field for inferring the fitness landscape of proteins.

Inspired by these tools, Hie et al.[20] showed that language models trained on a set of evolutionarily related sequences are capable of predicting the potential risks of SARS-COV-2 variants with multiple mutations, and Karim et al.[22] further combined the language model score with structural modeling to monitor the risks of existing variants. These computational tools can work as high-risk variant monitors and help us predict the risks of emerging variants. However, as these methods focus on prediction and rely on existing data, they do not provide detailed views for 'perspective' variants and antigenic evolutionary potential. Taft et al.[21] performed deep learning on the RBM sequences and built a predictive profile for the variants in ACE2 binding and antibody escape for class 1, 2, and 3 antibodies. The proposed framework works quite well in finding prospective mutations, but they still have limitations: the mutations are found by brute-force search, so they only focused on a small subset of the RBD region, missed a large part of the Class 3 antibody epitopes, and did not take the class 4 antibodies into consideration.

In this work, we presented the MLAEP, built upon the existing data and approaches to forecasting the combinatorial mutations in the entire RBD region that contains high antigenic evolutionary potential

and may occur in the future. We hypothesized that under high immune pressure, the virus would tend to escape the antibody neuralization over a short-term time scale, and therefore the forecasting problem transforms into a search problem: starting from an initial sequence, it searches for a variant sequence within some edit distance range that has an improved antibody escape potential without losing much ACE2-binding ability. With the DMS datasets that directly measure the binding affinity of RBD variants towards ACE2 and eight antibodies from four classes, we built a multi-task deep learning model that could simultaneously predict the binding specificity of the variants towards the ACE2 and antibodies. Furthermore, we used existing variants with their sampling date from the GISAID database to validate our hypothesis: we found a surprisingly high correlation between our model scores and the variants' sampling time (Spearman $r = 0.65$, $p < 1e$-308). Next, with our model as the scoring function, we used the genetic algorithm[31,32] to generate synthetic RBD variants with high ACE2 binding and antibody escape potential. Interestingly, the in silico directed evolution shares similar mutations with the adaptive evolution in immunocompromised COVID-19 patients[33–35] and newly emerging variants like XBB.1.5. Finally, we conducted in vitro neutralizing antibody binding assay to verify the ability of MLAEP to accurately forecast variants with high immune evasion potential.

## Results

### Overview of MLAEP

We first developed and trained a multi-task deep neural network model capable of predicting the variant RBD binding specificity towards the ACE2 and antibodies from four classes, as shown in Fig. 1. The model receives two inputs: the variant RBD sequences and the ACE2/antibody 3D structures, and outputs the binding specificities of the two inputs. The model is then trained with a multi-task objective function to predict the binding specificities of the variant sequences towards all targets simultaneously.

We fine-tuned the ESM-1b (evolutionary scale modeling) language model[29] for the sequence feature extraction. The model is pre-trained on ~27 million nature protein sequences in the UniRef50 database[36]. Fine-tuning the model has been proven to be effective for a broad range of downstream tasks, including biophysical properties prediction, structure prediction, and mutation effects prediction. With the ESM-1b model, the amino acid sequences are converted into a dense vector representation. For the ACE2/antibodies structures, we first transformed the 3D structures into graphs based on their contact maps and biophysical properties, then used the structured transformer[37] for the structural feature extraction. With the two models as feature extraction modules, we added nine parallel linear classification layers to learn the sequence to function mapping conditioned on the binding target structures (Fig. 1a). As we have multiple binding targets for the variants, we used a hard-parameter sharing scheme to perform multi-task learning, where all modules share the same parameters across all nine tasks. Then, we trained the entire framework in an end-to-end manner. Finally, the model learns how to predict binding specificity for ACE2 and eight antibodies. Given an input RBD variant sequence, our model outputs nine scores corresponding to the ACE2 and eight antibodies. We defined the average of eight antibody scores as the predicted antibody escaping potential.

Our key hypothesis is based on antigenic evolution: the future viral variants tend to have a higher antibody escaping potential without losing much ACE2-binding ability under high immune pressure. Thus, the antibody/ACE2-binding specificity learned by our model can be used to provide a meaningful direction in searching for novel variants that may cause future concern. Inspired by the progress in the machine learning-guided protein engineering field[26,27], we used the trained multi-task model as the scoring function (Fig. 1a), took the average prediction scores from all nine tasks as the fitness score, and used a modified genetic algorithm for searching for novel variants with

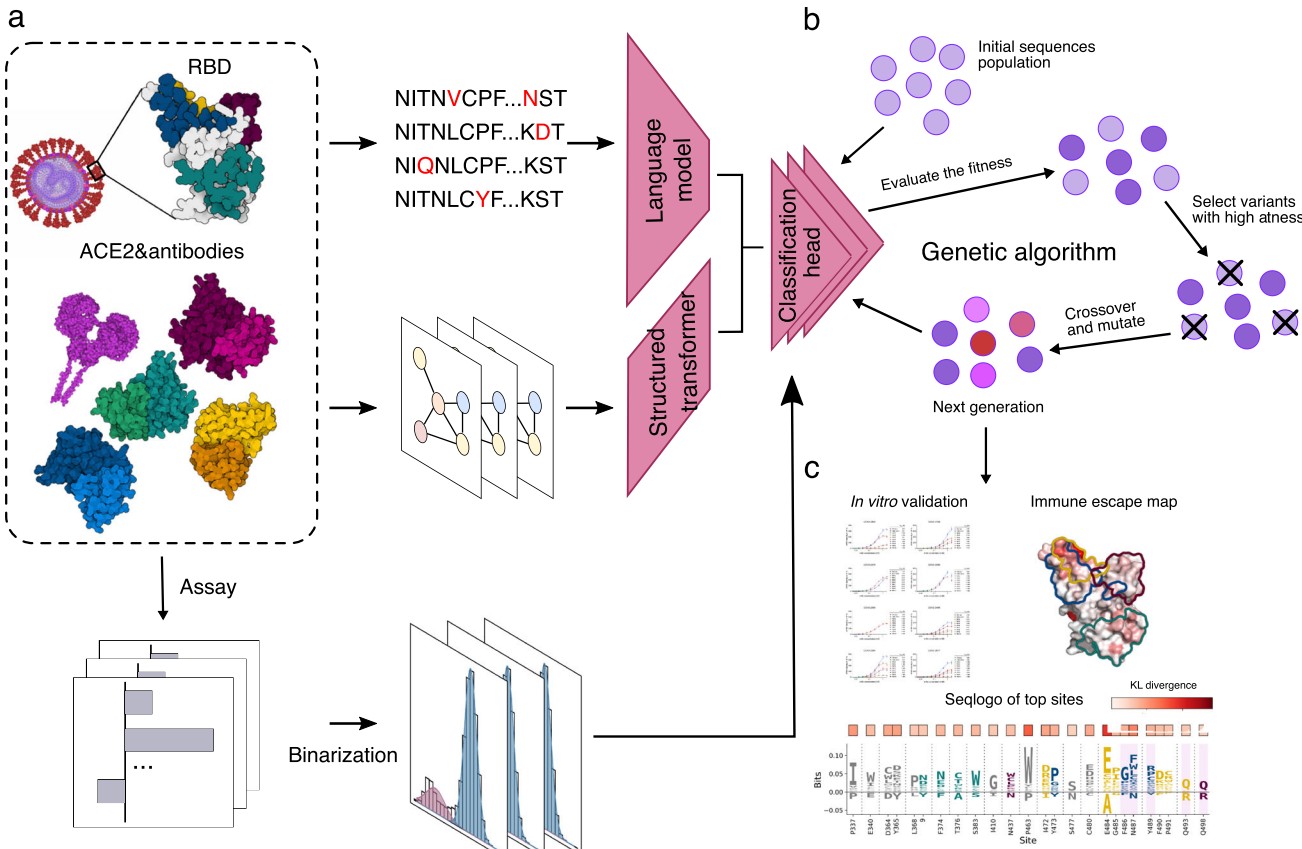

**Fig. 1 | Overview of the MLAEP framework. a** The multi-task learning model. We collected and cleaned the RBD variant sequences and their corresponding binding specificity to the ACE2 and eight antibodies. Then, the sequences and the structures of their binding partners were fed into the deep learning model with the multi-task learning objective. **b** The genetic algorithm. In silico-directed evolution was performed to navigate the virtual fitness landscape defined by the nine scores from the multi-task model. The generation loop was repeated multiple times until the desired functionality was reached. **c** These generated sequences were then subjected to validation experiments for evaluating their functional attributes.

improved fitness (Fig. 1b). The genetic algorithm is inspired by the process of natural selection, which iteratively evolves a group of candidates towards better fitness. The population of each iteration is called a generation. In each generation, the fitness of the candidate sequences is evaluated with the trained model. Then we filtered the populations by selecting the ones with higher fitness with higher probabilities for breeding the next generation (Fig. 1c). Random mutations and crossover are also introduced to better explore the search space. Genetic algorithm is known for performing well in solving combinatorial optimization problems, thus fitting our needs in searching for novel variants. More details can be found in Methods.

**The effectiveness of the multi-task learning model**

MLAEP follows the machine learning-guided directed evolution paradigm, while the quality of generated sequences largely depends on the sequence-to-function model. First, we validated the generalization ability of the models to newly seen variants with five-fold cross-validation. We collected and cleaned nine deep mutational datasets containing 19,132 variant sequences and their corresponding binding specificities towards ACE2 and eight antibodies from four functional classes. (Methods) We then compared a range of models specifically designed for protein engineering and assessed their classification performance in classifying the binders and non-binders (Methods, Supplementary Fig. 1) from the variant sequences, including the augmented Potts[30] model, the global UniRep[38] model, the eUniRep[26] model, the convolutional neural network (CNN), the long short memory neural network (LSTM), the recurrent neural network (RNN), the linear regression model, the support vector machine (SVM), the random forest and our model. The dataset is imbalanced regarding the number of positive and negative samples for all nine tasks. Thus, we reported the macro precision, macro recall, and macro-F1 score to add more weights to the minor classes. Combined with the structure features, our model outperforms the other advanced methods in predicting the effects of mutations in all nine tasks (Fig. 2a, Supplementary Figs. 2, 3, Supp Table 1). As a result, we focused on our model in the downstream analysis. We also performed an ablation study for our model to show the importance of each module. We found that both the fine-tuning step and the structure representations improves the overall model performance (Supplementary Fig. 4). We also conducted external validation experiments using several deep mutational scanning datasets[39,40] in addition to variant RBDs, and found that our model performed comparably and consistently well across all tasks (Supp Table 2).

To further validate the model's predictions for immune escape, we used the in vitro pseudovirus neutralization test (pVNT) datasets[41] that measured the cross-neutralizing effect of 17 RBD monoclonal antibodies against pseudoviruses expressing the Spike protein of selected variants of concern (VOCs). The pVNT assay reported the observed fold change in the $IC_{50}$ of the antibody response for these VOC-derived pseudoviruses, with lower fold change score indicating greater immune evasion compared to the wild-type (Wuhu-1) reference pseudovirus. Across all pseudoviruses and antibodies tested, we found surprisingly high correlations (Fig. 2b, Supplementary Fig. 5, Supp Table 3) between the predicted antibody escape potential and the log fold change in the $IC_{50}$.

The Evo-velocity[42] enables the inference of evolutionary dynamics for proteins with a deep learning model. It was built upon the premise

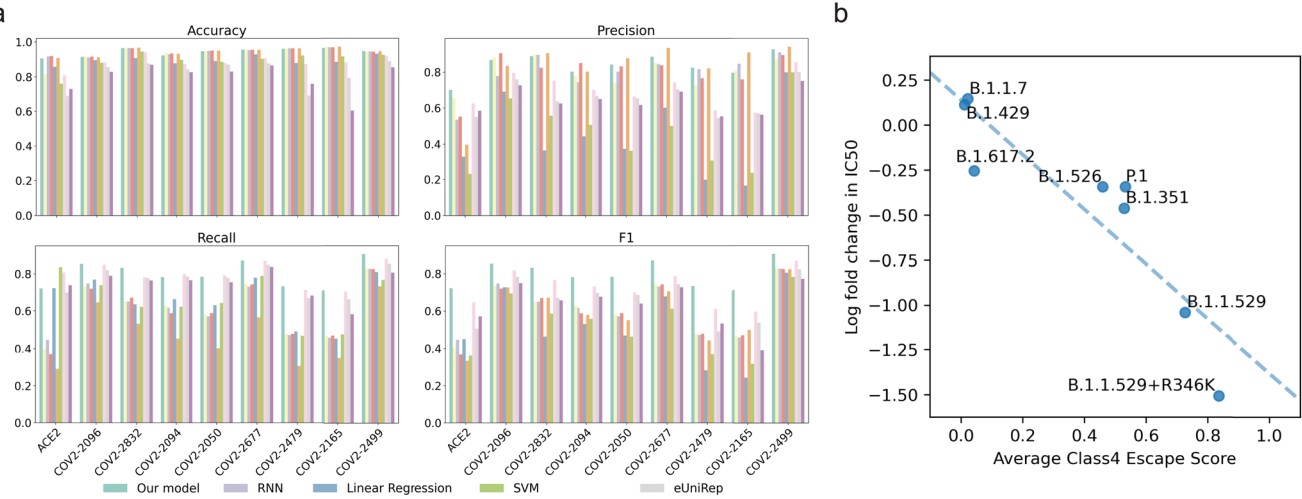

**Fig. 2 | Performance evaluation and in vitro pVNT experimental data validation. a** Model performance comparison for the classification of ACE2 and antibody binding specificity across different algorithms. Including our model, augmented Potts model, eUniRep model, gUniRep model, CNN, RNN, LSTM, linear regression, SVM, and random forest. The details of model implementation are given in Methods and performance metrics were calculated according to the equations provided in the Methods. **b** Validation of the predicted immune escape potential using the class 4 monoclonal antibody-based pVNT assay data (Antibody 10–40). The *x* axis indicates the model predicted variant escape potential, while the *y* axis is the log fold change of the VOCs compared with the wild type.

that global evolution occurs through local amino acid changes and leveraged protein language models to model the local rules of evolution (Methods). We next assessed our model's ability in inferring the evolutionary trajectory of the existing RBD sequences using the Evo-velocity. We used the existing SARS-COV-2 RBD sequences from the GISAID database across a time scale of around 27 months, from Dec. 2019 to Mar. 2022. The existing GISAID variant sequences were first transformed into embeddings with our multi-task model. On top of the embeddings, we assigned directions among them based on the changes in the average score predicted by our model, which forms the evolutionary "vector field". We visualized the embeddings in the two-dimensional space with the Uniform Manifold Approximation and Projection (UMAP)[43] (Methods). The variants of concern, including Alpha, Beta, Delta, and Omicron, were mapped into different clusters, and the velocities among these variants matched well with the known evolutionary trajectory (Fig. 3a). Despite the model being trained only with the RBD sequences, the pseudo time inferred with our model had a Spearman correlation of 0.55 ($p < 1e$-308) with the known variant sampling time (Supplementary Fig. 6). While using the ESM-1b (the Evo-velocity default setting) model, the score dropped to −0.38 ($p = 1.05e$-243) quickly (Supplementary Fig. 7). We noted that a large set of mutations occur outside the RBD region; this may explain the weak correlation between the ESM-1b model pseudo time and the sampling time. Longer sequence length (e.g., using the entire Spike protein region) would lead to better performance for the ESM-1b model[42]. We attempted to explain our model's unique ability to infer pseudo time with only the RBD region. We explored the effectiveness of labels in our supervised learning, as it provides alternative directions rather than the language model preference[42,44]. Interestingly, we found that the model prediction scores alone have an even higher Spearman correlation score of 0.65 ($p < 1e$-308) with the sampling time (Fig. 3b) compared with that of the inferred pseudo time, while for the predicted antibody escape potential, the Spearman correlation is 0.67 ($p < 1e$-308). These findings verify our assumptions: under the immune selection pressure, the virus evolves in the direction of immune escape, and our model can capture the antibody escape potential of the viral variants.

We next assessed the antigenic evolution on a short time scale by comparing the model predictions against the sampling time (Fig. 3c, Supplementary Fig. 8). We evaluated three types of scores, the ACE2-binding score, the antibody escape potential, and the weighted average of the two scores. The predictiveness of the antibody escapes score increases from nearly non-informative early in the pandemic to a stronger correlation during the Omicron wave. It also gains predictiveness with the emergence and spread of Alpha variants in Early 2021 but subsequently loses the predictiveness along with the emergence of other variants. We noted that the antigenic evolution For the ACE2-binding probability score, it tends to become more informative during the first year, while soon it becomes non-informative when the new VOC like Delta and Omicron emerged. These results suggests that the antigenic evolution happens along with the infection waves.

We then examined the model sequence representations against the binding specificities. We found that after the training, there are strong correlations between embeddings' primary and secondary axis of variation and the binding specificities for all nine targets (Fig. 3d, Supplementary Fig. 9). The correlations are observed for both ACE2 binding and antibody escape, suggesting that our multi-task learning strategies enable the model to learn the functional properties simultaneously. Given that the variant sequence embeddings are shared across tasks, this suggests that our model split the sequences based on an antigenic meaningful sense of binding preference.

In summary, our model effectively infers the immune escape potential and the ACE2-binding specificity, while the predicted scores correlate positively with the real-world sampling time, especially for the newly emerging Omicron wave. Taken together, we hypothesize that our model can work as a good scoring function for searching for high-risk mutations and the corresponding variants.

**In silico-directed evolution as a predictive tool**
With our model as the scoring function, we used the genetic algorithm to search for novel RBD variant sequences with high antigenic evolutionary potential. The search process consists of selecting an initial sequence from the GISAID database, generating and selecting "better-than-initial" sequences with the genetic algorithm to produce 38,870 putatively high-risk variants within a 15 mutations "trust radius" of the initial sequence (Methods). We performed the search process for the sequences in the GISAID database from 1 January 2022 to 8 March 2022, yielding a total of 971 distinct sequences. We then visualized the generated sequences together with the existing sequences using the distance-preserving multidimensional scaling plot[45] (Fig. 4a). While the

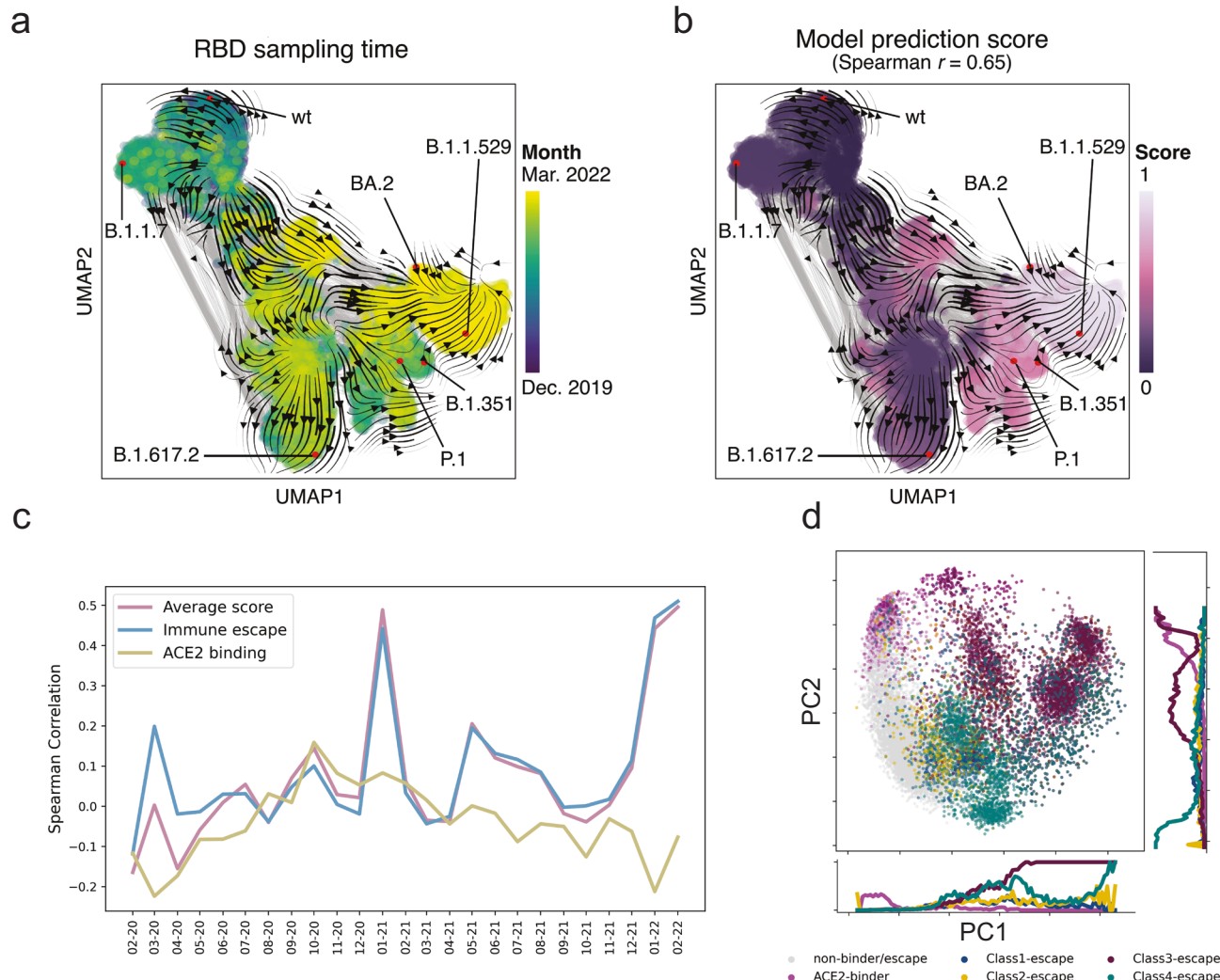

**Fig. 3 | Multi-task model captures the antigenic evolutionary potential. a** The landscape of SARS-COV-2 RBD variant sequences (obtained from GISAID), represented as a KNN-similarity graph (with the darker blue region represents less recent date, e.g., 2019, and yellow represents more recent date, e.g., 2022). The gray lines indicate graph edges, while the colored points are sequences with the known sampling time. The streamlines among the points show a visual correlation between model-predicted scores and the known sampling time. **b** Use the average score of our model to visualize the landscape. The landscape is colored by the model prediction score with darker colors represent lower scores and lighter colors represent higher scores. **c** Spearman correlation overtime for the model predictions, including the ACE2-binding score, immune escape potential, and the weighted average of the two in a time window of previous three months for each sampled date. (From February 2020 to February 2022) **d** Principal component analyses of the sequence's representations from our model, colored by the escaping/binding ability towards COV2-2832, COV2-2165(class 1 antibody), COV2-2479, COV2-2500 (class 2 antibody), COV2-2096, COV2-2499 (class 3 antibody), COV2-2677, COV2-2094 (class 4 antibody) and ACE2.

sequences from the deep mutational scanning experiments only occupy a small region around the wild type sequences, the prevalent variants (e.g., Omicron) locate in different regions, far from the wild type. The sequences searched with our model shown are diverse, largely expanding the sequence space.

Compared with the seed sequences, the synthetic sequences generated by our model include key mutations for ACE2 binding and antibody escape. To visualize the difference and further explore the patterns of the generated mutations, we constructed the position frequency matrix (PFM) for the two sequence sets and calculated the Kullback-Leibler divergence (KL divergence) for each position based on the two PFMs (Methods). Figure 4b and Supplementary Fig. 10 provides structure-based visualizations and projects the Kullback-Leibler divergence per site onto a crystal structure of the RBD (PDB id: 6m0j). As an alternative representation, Fig. 4c provides a probability-weighted Kullback-Leibler logo plot[46] for the top 50 most divergence sites, where the total height of the letters depicts the KL divergence of

the site, while the size of the letters is proportional to the relative log-odds score and observed probability (Methods). The logo plot for all positions can be found in Supplementary Fig. 11. Enriched amino acids locate at the positive side of the y-axis and depleted amino acids locate at the negative side.

The logo plot shows that the mutations searched by our model largely overlap with the antibody escape maps. For example, Y453, F456, and A475 are key sites for class 1 antibody escape[18], while they are also present in many synthetic variant sequences. Mutations escaped class 2 antibodies at sites E484, F490, and P491[18]. The logo plot shows that these sites ranked high as "active sites". Class 3 antibodies, which bind the opposite side of the receptor-binding motif, tend to be escaped by sites like N437, N448, and Q498[18], which are also vulnerable sites suggested by the model. Class 4 antibodies bind to a conserved motif among the sarbecorviues, far away from the RBM. Our model still captures the conservation and assigns mutations to the motif. However, some sites with a large KL divergence do not locate in the epitope

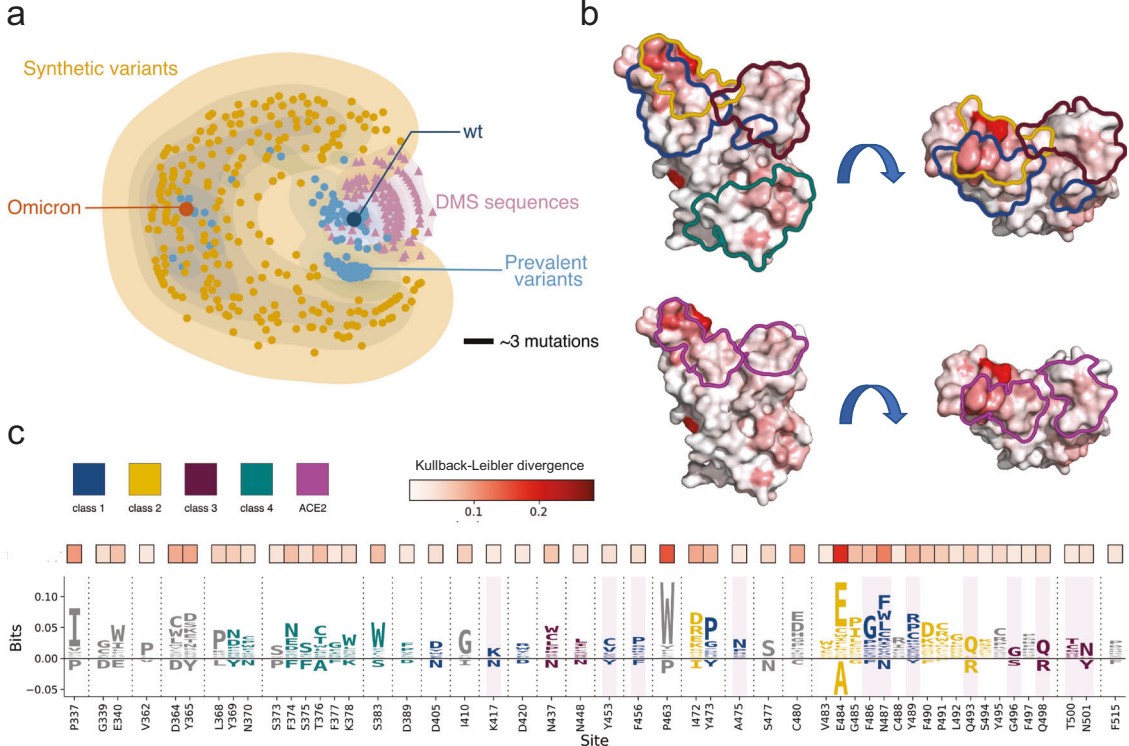

**Fig. 4 | Overview of the synthetic sequences. a** Distance-preserving multi-dimensional scaling plot illustrates synthetic sequences' diversity compared to existing variants and deep mutagenesis sequences. A scale bar of three mutations is shown. **b**, **c** the differences between the initial sequences and the synthetic sequences. **b** The surface of the RBD protein, colored by the KL divergence between the initial sequences and the synthetic sequences. Colored outlines indicate the epitope structural footprint. **c** The top 50 sites with the highest KL divergence value are selected for visualizing the difference between the generated sequences and the existing sequences. Enriched amino acids are located on the positive side of the *y* axis and depleted amino acids are located on the negative side.

regions. This has several explanations. Firstly, it is clear that some top sites (e.g., L368, C480) are not the direct binding sites but the proximal contact sites in the structures, which may influence the binding as well. Secondly, as there are epistasis relationships among the mutations, some combinatorial mutations may influence the RBD function non-linearly and then modify the antibody escape, which is not directly revealed by the epitope map. Moreover, these non-epitope sites with high KL divergence need to be taken into consideration as they may perform an important role in future variants. Another concern is that some sites in the epitope region have a low KL divergence, one possible explanation is that these sites have no tolerate mutations, for example, G416 and R457. Another explanation is that some mutations at antibody-contact sites do not directly influence antibody binding.

The synthetic variant sequences share similar mutations with the chronic SARS-COV-2 infections. A reverse mutation, R493Q, for example, was found in a persistently infected, immunocompromised individual[47]. Other mutations found by our model, like E340K[48], E484T[33], G485R[49], and F490L/E484G[50], are also found in immunocompromised patients treated with monoclonal antibodies. Moreover, the unique mutations found in the emerging variants, BA.4/5, the L452R, F486V, and the reverse mutation R493Q, are captured by our model. For the newly emerging variants like XBB.1.5, the key mutation that lead to increased transmissibility and immune escape, F486P[51], is also captured by our model (Supplementary Fig. 11). This suggests that our model could be used for finding novel mutations that may occur naturally. A detailed list of the found mutations in compromised patients is available in Supp Table 4. We next evaluated the immune evasion potential posed by the variant sequences using Evo-velocity analysis and viral language model risk inference, followed by structure modeling and antibody-antigen docking. The computational validation experiments suggest that the generated variants have high

immune escape potential. Further details can be found in the Supplementary Note 1.

## In vitro validation of novel mutations found by MLAEP

Having generated the synthetic sequences and found interesting single mutations, it is thus crucial to validate the risk and the immune evasion ability of combinatorial novel mutations using in vitro neutralizing antibody binding assay, especially for those that cannot be predicted with a linear additive model. Though the Omicron and its sub lineage are desired targets, they already exhibit high antibody escape abilities on the eight antibodies we selected for training our model, making it difficult to distinguish the effectiveness of novel mutations induced by MLAEP. To envision the differences, we used the RBD sequence of the Delta variant as the initial state and ran the entire framework again to generate and select "better-than-Delta" sequences. Our goal was to find possible antigenic evolutionary pathways for Delta that lead to high immune evasion.

We generated 3876 putatively high-risk variants using MLAEP and selected eight variants (Fig. 5, Supplementary Fig. 12) with unique immune evasion properties, including epistatic and non-epitope mutations. For example, the RBD3 contains seven mutations compared to the wild type, but all the single mutations are experimentally validated[18] to be ineffective at evading the eight antibodies we used. However, our model predicted that the RBD3 would have high immune evasion. The RBD4 does not contain mutations on the Class 4 antibody epitope, but our model predicted that it would escape Class 4 antibodies. The selection criteria are detailed in Supplementary Table 5.

We first expressed and purified the eight neutralizing monoclonal antibodies and ten RBDs (including wild type, Delta, and eight synthetic RBD we generated) bearing different mutations. We tested different combinations of neutralizing antibodies and RBD variants in a

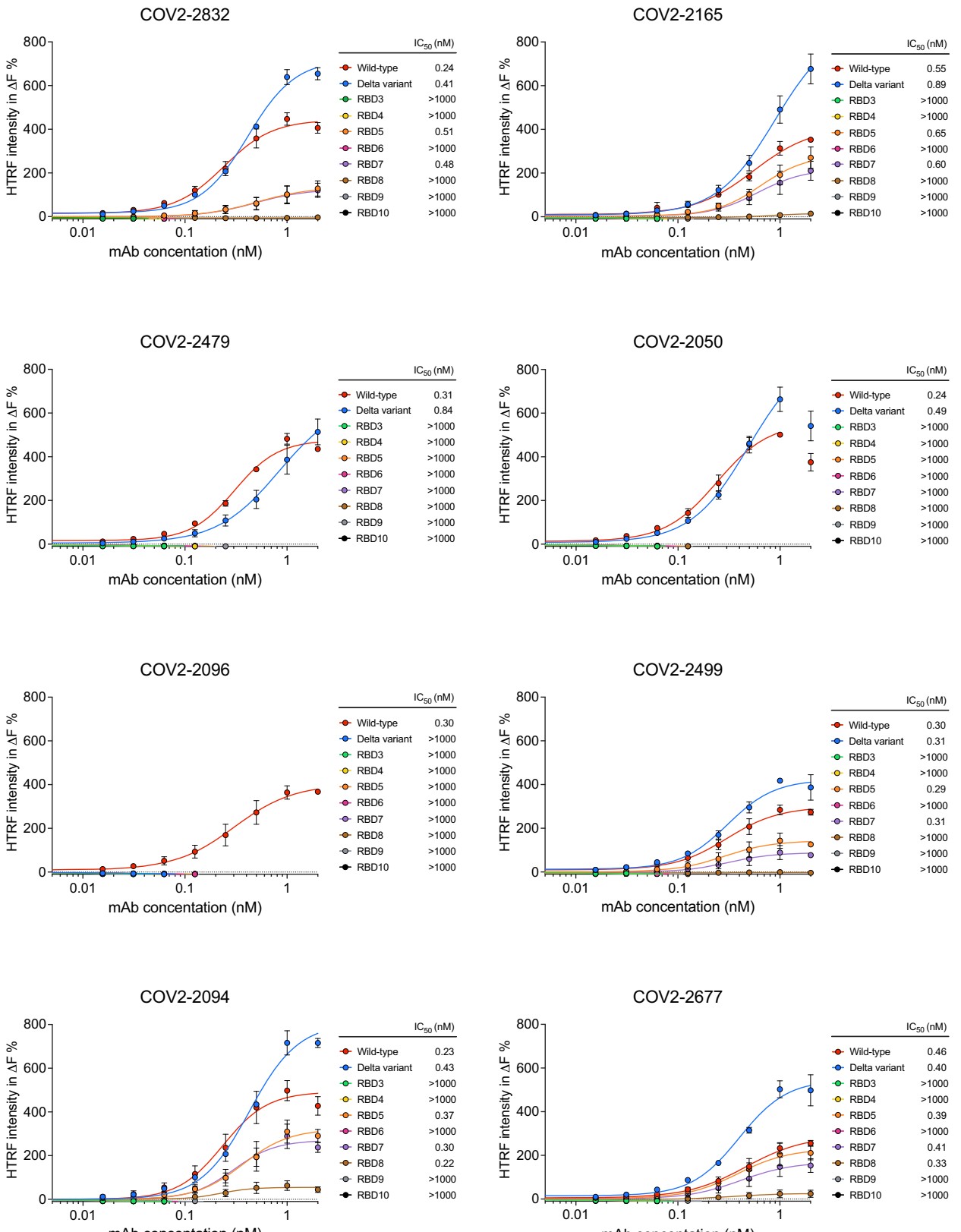

**Fig. 5 | Epitope mutations confer RBD resistance to the binding of neutralizing mAbs.** HTRF-based binding assay of wild-type and mutant RBD proteins against two representative anti-RBD monoclonal antibodies from four classes, including COV2-2832 and COV2-2165 (class 1 antibody), COV2-2479 and COV2-2050 (class 2 antibody), COV2-2096 and COV2-2499 (class 3 antibody), as well as COV2-2094 and COV2-2677 (class 4 antibody). ΔF% values were calculated from raw data and fit into dose–response curves, and the IC50 values were listed side by side. Data are presented as mean values ± standard deviation ($n$ = 3 independent experiments). Source data are provided as a Source Data file.

Homogeneous Time-Resolved Fluorescence (HTRF) based antigen-antibody binding assay. In our HTRF-based binding assay, the wild-type and Delta variant RBDs exhibited high binding efficacy against different neutralizing monoclonal antibodies, with the $IC_{50}$ falling in between 0.2 nM and 1 nM (Fig. 5). Notably, the Delta variant RBD showed no binding interaction to COV2-2096 (Fig. 5), consistent with the literature that the L452R[18] mutation on Delta variant confers evasion ability against this neutralizing antibody. Intriguingly, all our predicted synthetic variants exhibited reduced or diminished binding efficacy against all four classes of neutralizing antibodies targeting different epitope regions (Fig. 5). Specifically, RBD4, RBD7, RBD8, and RBD9 exhibited evasion or reduced binding to COV2-2094 and COV2-2677, two representative class 4 neutralizing monoclonal antibodies, even without bearing any mutations in the class 4 epitope region. We also found that RBD8 could completely escape class 3 antibodies (COV2-2096 and COV2-2499) without bearing mutations in the class 3 epitope region, suggesting that epistasis relationship play significant roles in the immune evasion, and such relationships could be captured by our deep learning model. The RBD5, RBD7, and RBD8 variants retained sensitivity to class 1 (COV2-2832, COV2-2165) and class 4 (COV2-2094, COV2-2677) antibodies with similar $IC_{50}$ values compared to wild-type RBD, but their binding efficacy to these neutralizing antibodies were reduced by large degrees. Overall, the synthetic variants and the novel combinatorial mutations generated from MLEAP exhibited a high potency for immune evasion, suggesting MLAEP captures the antigenic evolutionary potential.

## Discussion

In this paper, we proposed a machine learning-guided antigenic evolution prediction paradigm for forecasting the antigenic evolution of SARS-COV-2. We trained a multi-task deep learning model to predict ACE2/antibody binding specificity using variant sequences and binding target structures. Predicting ACE2 binding specificity is a relatively easy task, as one can capture the binding specificity using unsupervised learning-based models[29]. However, predicting antibody binding specificity is much more challenging and less explored in the literature. Through various validation experiments, we showed that our model can predict the antigenic evolutionary potential resulting from high immune pressure. Combined with the genetic algorithm, we conducted in silico-directed evolution using the model scores. The resulting synthetic sequences displayed high immune evasion potential, which we further validated using in silico computational tools and in vitro neutralizing antibody binding assay. MLAEP captures mutations that also happen in chronic SARS-COV-2 infections and emerging variants like BA.4/5 and XBB.1.5. In addition, MLAEP forecasts novel combinatorial mutations that affect antibody binding beyond epitope regions. While we used the genetic algorithm to search for novel variants, other search algorithms like hill-climbing[52], simulated annealing[53], and reinforcement learning[54] could also be combined with MLAEP. The multi-task learning model could be also replaced with other mutation effects prediction models[30].

Deep learning models can learn high-order epistasis relationships among the multiple mutations[28,29]. Our multi-task model, meanwhile, can capture such relationships and work as a monitor for predicting the escaping potential of newly emerging variants, particularly heavily mutated variants. Our in vitro HTRF-based high throughput assay verified that MLAEP is able to forecast epistatic and non-epitope mutations, thus expanding our understanding and ability to predict the virus evolution. When combined with the Evo-velocity analysis, our model helps to reveal the evolution trajectory of existing sequences and enables the discovery of high-risk variants that may appear in the future. The results suggest that the in silico-directed evolution can lead to the prediction of in vivo virus evolution. Consequently, MLAEP may enable the support of public health decision-making and guide the development of new vaccines. Besides, our approach could also be applied to rapidly evolving viruses and other potential outbreaks, such as antibiotic resistance[55].

An important property of MLAEP is that we focused on predicting the directionality of the mutation effect (i.e., whether a mutation increases or decreases binding affinity) rather than the magnitude of the effect. We plan to further develop our model to capture the quantitative effect of mutations in the future. Besides, one limitation of our model is that we only focused on the RBD sequences, while many mutations occur outside the region. We noted that the mutagenesis-assayed data provides semantically meaningful directions for finding "better-than-natural" sequences. An increasing number of experiments characterize the functionality of mutations in other regions, and we plan to explore these datasets in the future. Another concern is that we only optimized two targets, the ACE2 binding and antibodies escape, while the directionality of evolution is also driven by many other properties, like the epidemiology features and T-cell responses. In addition, the limited availability of variant ACE2 datasets prevented our model from capturing the full fitness landscape. Furthermore, the virus evolves continuously, making the set of effective neutralization antibodies change over time. Fortunately, the increasing availability of deep mutational scanning datasets[19,56] makes it convenient to track and update our model regularly. In the future, we will use these datasets and incorporate more in vivo and in vitro experimental data. Specifically, we will combine the in vivo antibody-antigen co-evolution data from patients and the assessment of other immune responses to better understand and predict the evolution of SARS-COV-2.

## Methods
### Dataset

We collected and cleaned nine deep mutational scanning datasets, which measure the binding affinity of the SARS-COV-2 RBD variants towards the ACE2 and eight antibodies from four classes. We built a dataset consisting of 19132 RBD sequences, where each sequence has nine labels, corresponding to their binding ability to the nine targets. Most sequences have one or two mutations compared to the wild-type RBD sequence. However, considering the possible batch effect and the physical meaning differences among the measured scores, we normalized each score independently by transforming the continuous variables into semantically meaningful binary labels. For the ACE2 task, we directly compared the binding score of the mutated sequences to the wild type and set the label to "enhanced binding" if the score is larger than the wild type and vice versa. There is no information about the wild type for the eight antibodies tasks, so we cannot set the threshold as described earlier. Instead, we found that the distributions of the binding score's logarithm clearly show two clusters; therefore, for all antibody datasets, we took it as a mixture-of-Gaussian model respectively, defining the one with smaller binding scores as escaped and vice versa. This preprocessing step is consistent with the subsequent work[57].

In summary, there were 1540 (8%) mutated RBD sequences identified as enhanced binding to ACE2, 3482 (18%) mutated RBD sequences identified as escaped to COV2-2096, 1220 (6%) mutated RBD sequences identified as escaped to COV2-2832, 2000 (10%) mutated RBD sequences identified as escaped to COV2-2094, 1473 (8%) mutated RBD sequences identified as escaped to COV2-2050, 1859 (10%) mutated RBD sequences identified as escaped to COV2-2677, 929 (5%) mutated RBD sequences identified as escaped to COV2-2479, 780 (4%) mutated RBD sequences identified as escaped to COV2-2165 and 3347 (17%) mutated RBD sequences identified as escaped to COV2-2499.

We used the pseudovirus neutralization test assay data from Liu et al.[41] to validate our model performance. The dataset measures the immune escaping of 10 high-risk variants pseudoviruses by comparing the fold change in $IC_{50}$ of 17 monoclonal neutralizing antibody responses against wild-type pseudovirus.

## Overview of the multi-task model

A central feature of SARS-COV-2 is antigenic evolution, that is, under high immune pressure, the newly emerging variants will tend to escape the antibody while do not lose much binding ability to the ACE2.

To accomplish the goal of predicting antigenic evolution, we need to construct the virtual fitness landscape of the antigenic regions, especially for the RBD protein. We aimed to infer the fitness landscape of the RBD by learning the effects of mutations on ACE2 binding and antibody escape. Specifically, given RBD variant sequences and their labels, together with the binding partner (ACE2/antibody) structures, our model learned the nonlinear mapping function $f$ that can simultaneously predict the binding specificity for ACE2 and antibody. The function $f$ is parameterized by learnable mapping parameters $\theta$ composed of three modules: the sequence feature extractor $\mathscr{S}$, the structure feature extractor $\mathscr{G}$ and the sets of nine classification heads $H = \{\mathscr{H}_c\}_{c=1}^9$, where all $\mathscr{H}_c$ share the same group of parameters. All three modules are neural networks. The parameters of the three modules are optimized in an end-to-end manner.

1. Sequence feature extractor. The sequence feature extractor takes as input of amino acid sequences of RBD variants $\mathbf{x} = (x_1, x_2, \ldots, x_l)$ of length $L$, where $L$ denotes the length of the RBD sequence and elements $x_i$ belongs to $A$ = {all amino acids}. Input is mapped to a dense representation vector (sequence representation). The backbone of the sequences feature extractor is the ESM-1b transformer, which is pretrained on UniRef50 representative sequences with the masked language modeling objective. We chose the ESM-1b as the sequence feature extractor because it outperforms other baselines on a range of downstream tasks. The pretrained weights were used for initializing the neural network, and we fine-tuned the model parameters during training.

2. Representing structure as graph. We first represented the 3D structure as a k-nearest neighbor graph $g = (V, E)$ with the node set $V = \{v_j\}_{j=1}^N$ of size $N$, where each element $v_j$ denotes for the features of representative atoms (we chose N, C, and O atom in the experiment) in the protein 3D structure, $N$ denotes the total number of atoms. For each atom, we got its two nearest neighbors with the following constraints: the ones with the same atom type but belong to different amino acids. We then measured the dihedral angles of the atom and its neighbors to as node features. The edge features $E = \{e_{ij}\}_{i \neq j}$ describes the relationship between the nodes, including the relative distance, direction, and orientation between the two nodes in the three-dimensional space. We set $k$ as 30.

3. Structure feature extractor. The structure feature extractor tasks as input of graphs $g = (V, E)$ describing the spatial feature of the protein structure. The transformed graph is further mapped into a dense representation vector (structure representation). The backbone of the structure feature extractor is a Structured Transformer[37], where the attention for each node is restricted to its k-nearest neighbors in 3D space. We chose the Structured Transformer for the structure feature extraction as it is computationally efficient and performs well in the protein design task. The structure representation works as conditional tags in our multi-task learning.

4. Classification heads. After getting the sequence representation and the structure representation, we concatenated the two vectors into the joint representation, and fed it into the classification heads. The classification heads map the joint representation to the labels. We used nine parallel classification heads for the nine classification tasks, while the neural network parameters are shared. During training, the sequence feature extractor, the structure feature extractor, and the classification heads are trained in an end-to-end manner to minimize the average classification loss among the nine tasks.

5. Loss function. Let $\mathbf{x} = \{x_i\}_{i=1}^N$ be the set of RBD variant amino acid sequences, and $\mathbf{y} = \{y_i\}_{i=1}^N$ be the set of labels of all sequences, and $\mathbf{y} = \{y_i\}_{i=1}^N$ denotes for the set of $M$ labels of the $i$-th RBD variant. Furthermore, let $G = \{g_c(V,E)\}_{c=1}^M$ consists of M graphs derived from the ACE/antibody structures. We seek to learn a joint embedding for all downstream classification tasks to better model the fitness landscape of RBD. Therefore, the sequence feature extractor and the structure feature extractor are shared among all tasks. Considering that all the tasks are imbalanced in terms of the positive and negative samples, we added a rescaling weight $p_c$ to all tasks and optimized the following loss function:

$$L = \frac{1}{MN} \sum_{c=1}^M \sum_{i=1}^N -[p_c y_i^c \cdot \log \sigma(\mathcal{H}_c(\mathcal{S}(x_i)\mathcal{G}(g_c))) \\ + (1 - y_i^c) \cdot \log(1 - \sigma(\mathcal{H}_c(\mathcal{S}(x_i)\mathcal{G}(g_c))))] \tag{1}$$

Where $p_c$ equals to the number of positive samples divided by the number of negative samples, M equals to nine, $\sigma$ is the sigmoid function. The equation measures the binary cross entropy between the targets and predicted probabilities.

## Architecture and hyperparameters

The architecture of the sequences feature extractor is based on the ESM-1b transformer, which consists of 34 layers, we used the outputs of the 33rd layer as the sequence feature representations. For the structured transformer, we only kept the transformer encoder, and used three layers of self-attention and position-wise feedforward modules with a hidden dimension of 128. Finally, we got a 1280-d vector for each sequence as the sequence representation and a 1300-d vector for each 3D structure as the structure representation. For each classification head, we used 1024 neurons in the first layer and two neurons in the second layer. The RELU function is used between the layers as nonlinear activations. We also passed a dropout rate $p = 0.5$ and added weight decay to prevent overfitting. We trained the entire model with the AdamW optimizer and used a linear schedule with warmup to adjust the learning rate. We set the batch size as 16 and gradient accumulation steps as 10, which means that the total train batch size is 160, and the validation is the same. We used a weighted random sampler function for our training batches, which oversamples the minority class to ensure that the number of samples in each class are equal or close to equal. The model was trained for 9500 updates with the initial learning rate of 1e-5 and warmup steps 120, during which the model with the best marco F1 score among all the tasks was kept. The hyperparameters described above were decided through several trials of experiments and selected the one with the best performance.

## Choice of baselines

Our framework follows the machine learning-guided directed evolution paradigm, while the quality of generated sequences largely depends on the sequence-function model. Thus, we validated the generalization ability of the models to newly seen variants with cross-validation. Our multi-task deep learning model was designed as a supervised technique to infer the fitness of variant sequences, while there is no existing method designed for multi-task learning or multi-label learning on this task. So instead, we evaluated the performance of the existing methods separately for all nine tasks. We chose the state-of-the-art supervised-learning-based methods for inferring the effect of mutations, including the augmented Potts model[30], the eUniRep model[26], and the gUniRep model[38]. We also benchmarked several baseline machine learning methods including CNN, LSTM, RNN, Linear Regression, SVM, and Random Forest.

The augmented Potts model combines the evolutionary information with the one-hot encoded amino acid sequences as input

features and trains a linear regression model on top of the features. It outperforms most existing methods in inferring the effects of mutations. We first generated the multiple sequence alignments profile of RBD using the profile HMM homology search tool Jackhmmer. We set the bit score threshold as 0.5 and the number of iterations to 1. We then calculated the evolutionary Potts potential of the RBD variant sequences using the plmc. We replaced the Ridge regression with a Logistic regression head for the classification objective while keeping the rest procedures the same as the original settings.

The gUniRep model was trained on 24 million UniRef50 amino acid sequences with the next amino acid prediction objective, and the representations extracted from the pretrained model acts as a featurization of the sequences, benefits the downstream protein informatics tasks. With the RBD variant sequences as input, we got the fixed-length vector representations from the pretrained model as sequence embeddings. We added a Logistic regression head for downstream classification.

The eUniRep model was built on top of the gUniRep model. An unsupervised fine-tuning step with sequences related to the target protein (evotuning) was introduced to learn the distinct features of the target family. Previous in vitro studies on the GFP and beta-lactamase proved its effectiveness for efficiently modeling the protein fitness landscape. We performed evotuning with the same MSA profile we generated in the augmented Potts model. After that, we characterized the sequence embeddings with the eUniRep model and train a Logistic regression model for downstream classification. All methods were trained and tested on the same training data and validation data for all five folds.

The CNN model consisted of a feature module and a classification module, where the feature module was composed of two 1D convolution layers, max-pooling layers, and ReLU layers. A 128-dimensional feature vector generated by the feature module was used to predict the label by the classification layer.

The LSTM model consisted of a feature module and a classification module, where the feature module was composed of one 1D LSTM layers and two linear layers, followed a ReLU layer and a sigmoid layer. A 128-dimensional feature vector generated by the feature module was used to predict the label by the classification layer. The RNN model is similar to LSTM model, except replace the LSTM module with the RNN module. The Linear regression, SVM and Random Forest were implemented using Scikit-learn v1.1.0[58].

## Ablation study

We performed ablation studies to show the effectiveness of each module. We first explored the effectiveness of the graphical representations. We replaced the structure features with the random Uniform noise $X \sim U(0,1)$ and performed the multi-task learning with the same training details and procedures. Besides, we also explored the importance of fine-tuning by freezing the parameters of the ESM-1b model and only optimized the parameters of the classification head.

## Performance evaluation

We evaluated the multi-task learning model with the fivefold cross-validation. We randomly split the dataset into five folds. Each time, we used four folds as the training data and held out the remaining fold for validation. We used the Accuracy, Precision, Recall, and F1 score to evaluate the classification performance across all models. As all nine classification tasks are imbalanced, we used the Marco-precision, Marco-recall, and Macro-F1-score to get an unbiased evaluation. True positive (TP), true negatives (TN), false positives (FP), and false negatives (FN) were measured by comparison between the prediction results produced by the model and the ground truth in the validation set.

$$Accuracy = \frac{TP + TN}{TP + TN + FP + FN} \qquad (2)$$

$$Recall_c = \frac{TP}{TP + FN}, Recall_{macro} = \frac{Recall_1 + Recall_2}{2} \qquad (3)$$

$$Precision_c = \frac{TP}{TP + FP}, Precision_{macro} = \frac{Precision_1 + Precision_2}{2} \qquad (4)$$

$$F1_{macro} = 2 \times \frac{Precision_{macro} \times Recall_{macro}}{Precision_{macro} + Recall_{macro}} \qquad (5)$$

## Generate virtual RBD variant sequences with the genetic algorithm

To forecast the variants that follows the antigenic evolutionary potential, we applied the genetic algorithm for searching the peaks of the fitness landscape described by our model. Inspired by Darwin's theory of natural evolution, the genetic algorithm mimics the evolutionary process in the genome, where mutations, crossover, and selection happen, letting candidate solutions of a population with higher fitness scores have a higher probability of surviving and producing the next generation of offspring. For SARS-CoV-2, it has been proven that similar progress happens in immunocompromised infected patients who got treated with the monocle antibodies[33]. Hence, we used the genetic algorithm to model the antigenic evolution process and search for the potential risky variants that might appear in the future. The genetic algorithm we used consists of the following steps:

1. Selection of initial sequences. For the Omicron related experiments, the initial input sequences were obtained from the March 8, 2022 GISAID release[1]. We selected the RBD sequences from 1 January 2022 to 8 March 2022, resulting in 957 distinct RBD sequences. For the Delta experiments, the initial RBD sequence is the Delta variant RBD sequence. For each sequence, we created a generation $P_0$ of size $S$ by perturbing the sequences $S$ times to generate a set of distinct modifications to the original sequence.

2. Perturb operation. For a given sequence $\mathbf{x} = (x_1, x_2, \ldots, x_n)$, we first randomly selected an amino acid $x_i$, and got the $K$ nearest neighbors of the selected amino acid according to the BLUSUM62 matrix. Secondly, we computed the fitness value when $x_i$ is replaced with its neighbors, while keeping the remaining set of words unchanged. We then picked the mutation with a probability proportional to its fitness value. Finally, the selected mutated amino acid replaced the original one, we got a new sequence. We set $K = 20$ in our experiments.

3. Estimation of the fitness. The fitness score is defined as the average value of target label prediction probabilities for all nine tasks. For ACE2 binding task, the target label is binding, while for the antibody task, the target label is escape. The probabilities were found by querying the trained multi-task model.

4. Crossover. After getting the perturbed population and the fitness values for each individual sequence, we performed the crossover operation. Pairs of sequences are randomly selected with the probabilities proportional to its fitness value. A child sequence is then generated by independently sampling from the two parents. The newly generated sequences form the new generation. If the fitness value of a population member in the generation is higher than the high-risk threshold, the optimization is done. Otherwise, the perturbation, selection, and crossover operation will be applied to the new generation.

We performed in silico evolution for each initial sequence from GISAID subset for 100 times independently, and finally got 38,870 unique RBD variant sequences for the experiment. We obtained 3,876 unique RBD variant sequences derived from the Delta variant, which eight candidates were utilized in the HTRF-based neutralizing antibody binding assay (Supp Table 5) for the Delta-focused study.

### Evo-velocity analysis

The Evo-velocity analysis follows the study of Hie et al.[42]. They used the pretrained protein language model (e.g., ESM-1b) to predict the local evolution within protein families and used a dynamic "vector field" to visualize it. It involves embedding the sequences of interest as vectors in a high-dimensional latent space, where the geometric distance between the representation of proteins correlates with their actual structural, functional, and evolutionary relatedness. The evo-velocity between two sequences is calculated by considering the log-pseudolikelihood of observing a mutation from one sequence to another, providing a local mutational likelihood gradient around a particular protein. When looked at globally, this vector field gives insight into the directionality of the evolutionary process and can model global evolution. We first computed the embeddings for each sequence with the ESM-1b model, and then constructed the K-nearest-neighbor graph based on the embeddings, in which node represents the sequences and edges connected similar sequences. Further, the edges were assigned with directions based on the language model pseudolikelihoods, with flow-in node meaning evolutionarily favorable. Here, we performed the Evo-velocity analysis with their settings and ours. In our setting, we used the joint embeddings extracted from the fine-tuned protein language model and the Structured Transformer model to represent the sequences and set the direction of the edge by comparing the average predicted score among the nine tasks, where vertex with a large value is defined as the tail. We collected 7594 unique RBD sequences from the 8 March 2022 GISAID release. The date of the sequence is defined as the first reported date. After constructing the directed KNN neighborhood graph, we further performed network diffusion analysis to infer the pseudo time. We manually set the root as the wild-type RBD sequences.

### Visualization

We visualized the model embeddings using UMAP. The K-nearest-neighbor network was built with the $k$ set to 30, while the resolution is set to 1. We calculated the KNN graph and performed UMAP for both the GISAID sequences and the generated sequences. We further projected the predicted KL divergence maps onto a RBD structure (PDB id: 6m0j) and visualized the structure with PyMol. We collected the binding epitopes for class 1, 2, and 3 antibodies from Greaney et al.[18], while for the class 4 antibodies, we used the contact sites of antibody CR3022 to represent the class 4 binding epitope.

We used probability-weighted Kullback-Leibler Logo plot for visualizing the generated mutations. Let $M_1 = (\mathbf{f_1}, \mathbf{f_2}, \mathbf{f_3}, \ldots, \mathbf{f_n})$ denote the position frequency matrix (PFM) of the initial sequences, where the length of the initial sequences are $n$ and each $\mathbf{f}_i = (a_1, a_2, \ldots, a_{20})^T$, represents the frequency of each amino acid at position $i$. Further, let $M_2 = (\mathbf{f'_1}, \mathbf{f'_2}, \mathbf{f'_3}, \ldots, \mathbf{f'_n})$ denotes for the PFM of the generated sequences, each $\mathbf{f'_i} = (a'_1, a'_2, \ldots, a'_{20})^T$. We computed the KL divergence for each position:

$$D_{KL}(\mathbf{f'_i} || \mathbf{f_i}) = \sum_{i=1}^{20} a'_i \bullet \ln(a'_i / a_i) \tag{6}$$

The KL divergence denotes for the total heights at each position in the logo plot. We further set the height and the direction of a letter with a probability-weighted normalization[46], where the relative height

of each individual amino acid is proportional to $a'_i \bullet \ln(a'_i / a_i)$:

$$h(a'_i) = \frac{a'_i \bullet \ln(a'_i / a_i)}{\sum_{i=1}^{20} a'_i \bullet |\ln(a'_i / a_i)|} D_{KL}(\mathbf{f'_i} || \mathbf{f_i}) \tag{7}$$

### Antibody-antigen docking

We collected the crystal structures of representative antibodies of four antigenic classes from PDB database: LY-CoV16 (class 1, PDB id: 7c01), LY-CoV555 (class 2, PDB id: 7kMG), S309 (class 3, PDB id: 7R6W) and CR3022 (class 4, PDB id: 6w41). Next, we used MLAEP to generate risky RBD sequences based on the GISAID subset (1 January 2022 to 8 March 2022). The top 20 RBD sequences with highest score were selected and modeled by the protein structure homology-modeling server SWISS-MODEL[59]. Besides, we also collected the RBD structure of the Wuhan-wild type and Omicron variant. After that, the in silico docking simulation between RBD structures and antibodies was implemented with the Rosetta antibody-antigen docking protocols[60]. We applied the all-atom relax protocol, docking prepack protocol, and antibody-antigen docking simulation using the SnugDock with the complex structures of the combination of 22 RBD structures and four antibodies. We ran the docking 1000 times independently for each antibody-antigen pair and got 1000 Rosetta interface scores as binding scores. The Rosetta interface score ($I_{sc}$) is defined as

$$I_{sc} = E_{bound} - E_{unbound} \tag{8}$$

where $E_{bound}$ is the score of the bound complex, while the $E_{unbound}$ is the sum of the scores of individual docking partners. We further filtered out the scores with $I_{sc} > 0$ for getting reasonable scores. The statistical significance was tested using the two-sided $t$-test for the means of two independent samples of scores.

### Recombinant monoclonal antibody and RBD variants purification

The sequences coding SARS-COV-2 monoclonal antibodies were kindly provided by Prof. James E. Crowe from Vanderbilt University Medical Center. The LH and HC sequences were codon optimized and submitted to Genescript for custom human IgG1 antibody expression. Sequences of wild-type, delta variant, and synthetic variant RBD proteins were codon optimized and submitted to Twist for vector construction. All RBD constructs contain a secretion signal on the N-terminal, and a 6× his-tag followed by a strep-tag II on the C-terminal. In brief, Expi293 cells were transfected in 40 mL Expi293 Expression Medium (Thermo Fisher A1435101) at 37 °C, 8% $CO_2$ on an orbital shaker at 120 rpm. After five days, cells were removed by spinning at $500 \times g$ for 5 mins at 4 °C, and the medium was further centrifuged at $16,000 \times g$ for 5 mins at 4 °C. The supernatant was then mixed with his-tag purification resin (Beyotime P2221) on a shaker at 4 °C. After 1 hour of incubation, the mixture was loaded on a gravity chromatography column and washed for 15 mL of washing buffer [25 mM Tris, pH 8, 300 mM NaCl, and 1 mM DTT]. The elution was collected in 5 mL and loaded on another 2 mL column pre-packed with 0.5 mL Strep-Tactin XT 4Flow high-capacity resin (IBA Lifesciences 2-5030-025). The RBD proteins were eluted in 5 mL of washing buffer supplemented with 50 mM Biotin. For some mutant RBD proteins that have reduced secretion into the medium, cell lysates were prepared in lysis buffer [25 mM Tris, pH 8, 300 mM NaCl, 0.5% Triton X-100, 1 mM DTT, 1× protease inhibitor cocktail (PIC)] for 30 min on a shaker at 4 °C. Clarified lysates were subject to two affinity columns following the same purification protocols. All purified RBD proteins were buffer exchanged and concentrated to 1 μM in 1× PBS using Amicon, flash-frozen in liquid nitrogen, and stored at −80 °C.

## Homogeneous time-resolved fluorescence (HTRF) antigen-antibody binding assay

The binding intensity between purified SARS-COV-2 RBDs and neutralizing antibodies was measured as HTRF signals in the antigen-antibody binding assay. The HTRF donor and acceptor pair were chosen to target the his-tagged RBD proteins and human IgG1 antibodies, respectively. Briefly, a total of 10 µL reaction was set up on each well of the black, round-bottom, low-volume 384-well plates (Corning 4511) containing 5 nM purified wild type or mutant RBDs, 3 nM goat anti-human IgG conjugated with Alex Fluor 647 (Thermo Fisher A-21445), 0.33 nM monoclonal antibody anti-6His-Tb-cryptate Gold (Cisbio 61HI2TLA) and two-fold dilutions of neutralizing mAbs from 2 nM to 0.0156 nM in 1× PBS supplemented with 0.1% BSA, and 0.1% Tween-20. The plate was sealed with plastic film and incubated at room temperature for 1 hour. The HTRF signals were measured in CLARIOstar Plus (BMG LABTECH) with the excitation filter at 340 nm and the emission filters at 620 nm and 665 nm. The reading lag time and integration time were set to 60 µs and 200 µs, respectively. The HTRF ratios from samples and negative controls were calculated by dividing the intensity readouts from the 665 nm channel over the 620 nm channel. All ratios were background subtracted and normalized in ΔF%:

$$\triangle F\% = \frac{\text{HTRF ratio(sample)} - \text{HTRF ratio(negative control)}}{\text{HTRF ratio(negative control)}} \times 100$$

(9)

The $IC_{50}$ value was calculated by fitting the data into a dose-response curve in Prism 9. Data points with the 'hook' effect were removed from the fitting.

### Reporting summary

Further information on research design is available in the Nature Portfolio Reporting Summary linked to this article.

## Data availability

The deep mutational scanning datasets is publicly available at https://github.com/jbloomlab/SARS-CoV-2-RBD_DMS/blob/master/results/binding_Kds/binding_Kds.csv and https://media.githubusercontent.com/media/jbloomlab/SARS-CoV-2-RBD_MAP_Crowe_antibodies/master/results/escape_scores/scores.csv. The pseudovirus neutralization test assay data is publicly available in https://www.nature.com/articles/s41586-021-04388-0/figures/4. The prevalent hCoV-19 RBD sequences used in this study are based on metadata associated with 5,483,918 sequences available on GISAID up to March 8th, 2022, and accessible at https://doi.org/10.55876/gis8.230510mg. The PDB data was used for visualization and docking experiments, we used: PDB id: 6m0j; PDB id: 7c01; PDB id: 7kMG; PDB id: 7R6W; PDB id: 6w41. The generated variant sequences and other source data are provided as a Source Data file. Source data are provided with this paper.

## Code availability

We compared our multi-task model with gUnirep(https://github.com/churchlab/UniRep), eUnirep(https://github.com/churchlab/UniRep), and augmented potts model(https://github.com/chloechsu/combining-evolutionary-and-assay-labelled-data), following their github repository. We performed Evo-velocity analysis with https://evolocity.readthedocs.io/en/latest/. Besides, we visualized our model embeddings with UMAP version 0.5. We performed docking experiments with SnugDock in Rosetta 3. We visualized the protein structures with PyMol 2.4. We use Sklearn version 1.1.1 and Scipy[61] 1.6.0 for measuring model performance. The source code for this study can be accessed at the GitHub repository: https://github.com/WHan-alter/

MLAEP. The webserver can be found at https://mlaep.cbrc.kaust.edu.sa/. A permanent archive of the source code is also available on Zenodo at https://doi.org/10.5281/zenodo.7781867[62].

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

## Acknowledgements

We gratefully acknowledge the support of the King Abdullah University of Science and Technology (KAUST). [FCC/1/1976-44-01, FCC/1/1976-45-01, URF/1/4663-01-01, REI/1/5202-01-01, REI/1/4940-01-01, REI/1/5234-01-01, REI/1/5414-01-01 and RGC/3/4816-01-01] to X.G.; University Grants Committee's Collaborative Research Fund (C6036-21GF) to P.P.H.C; The Chinese University of Hong Kong's Research Committee Research Fellowship to X.X. We would like to acknowledge Li Ka Shing Translational Omics Platform (LKSTOP) for equipment support.

## Author contributions

W.H., C.N., X.X., P.P.H.C, and X.G. conceptualized the study and developed the methodology. W.H. and C.N. implemented models and analyzed data. X.X., P.P.H.C, R.Z., Y.W., and S.S. designed and performed the wet-lab experiments. A.S. developed the web server. Z.L., H.Z., and J.Z. helped with the baseline experiments. P.P.H.C and X.G. supervised the research and the entire project. All authors wrote the paper.

## Competing interests

The authors declare no competing interests.
