## [Peer Review File · Nature Communications]

Reviewer comments, first round review

Reviewer #1 (Remarks to the Author):

The manuscript by Han et al. describes that they develop Machine Learning-guided Antigenic Evolution Prediction (MLAEP) for exploring the antigenic evolution of SARS-CoV-2 spike protein. They train a multi-task deep learning model to predict ACE2- and neutralizing antibody-binding specificity of the spike variants. By utilizing in vitro neutralization dataset, they validate the model performance in predicting the escape potential. Moreover, MLAEP shows a strong correlation between variants' antigenic evolution and their sampling time. By generating the genetic algorithm-based synthetic RBD variants with high ACE2 binding and antibody escape potential, they find that in silico directed evolution shares similar mutations with the adaptive evolution in immunocompromised COVID-19 patients and new Omicron subvariants BA.4/5. Despite the establishment of the novel in silico model, this study raises serious concerns as follows;

Major points:

1. In general, such an in silico model should be supported by biological evidence. In this sense, whether this model could be properly validated in predicting the potential of antibody evasion by the spike proteins is largely dependent on the effectiveness of the in vitro pseudovirus neutralization test (pVNT) assay datasets, the paper of which is, however, a preprint <https://www.biorxiv.org/content/10.1101/2021.12.24.474095v1> and not certified by peer-review. Notably, this pVNT assays were conducted by using only a small number of sera obtained from individuals after the second dose of the BNT162b2 in the phase-I/II trial, which cannot reflect the current real-world data with booster shots. For this reason, the validation should be rather performed with the dataset of monoclonal antibody-based neutralizing assays.
2. The following sentence (Page 12, lines 3-4) "All of them (= CR3022, CB6/LY-CoV016, LY-CoV555, and S309 monoclonal antibodies) have been proven to be effective in neutralizing coronavirus (48-51)" is incorrect. Indeed, the papers cited here were published before the emergence of Delta. The truth is; the class 2 antibody LY-CoV555 is not effective against both Delta and Omicron, and the class 1 antibody CB6/LY-CoV016 is no longer effective against Omicron BA.1. The class 3 antibody S309, which is effective to Delta and Omicron BA.1, shows reduced neutralization activity to BA.2 so that it is currently unauthorized to treat COVID-19 in the USA. Based on these facts, LY-CoV555 and CB6/LY-CoV016 should show the higher binding scores than those of S309, unlike the results shown in Fig. 5d and 5e.
3. No supplementary tables (1, 2, 3, and 4) are available.

Minor points

1. No page and row numbers make it difficult to direct specific comments.
2. Abstract, lines 7 and 9: MLEAP -> MLAEP
3. Page 4, line 22: Joseph et al. -> Taft et al.
4. Page 5, Results "Overview of MLAEP": It is difficult to follow this multi-step model (Fig. 1) from the text. Please label figures as 1a, 1b, 1c...
5. Page 7, line 19: SARS-CoV-2-S pseudoviral -> SARS-CoV-2-spike pseudoviruses
6. Page 7, line 21: VOCs pseudo viral -> VOC-derived spike pseudoviruses
7. Page 8, line 19: Fig. 3c -> Fig. 3b
8. Page 12, line 4: It is unclear which 20 synthetic variants were used (described in a supplementary table?).

Reviewer #2 (Remarks to the Author):

This work adds to a number of recent reports that aim at predicting the evolution of SARS-CoV-2 during the pandemic. Previous publications use epidemiological data, deep mutagenesis scanning (DMS), language models and others. Here, the work appears to use a pre-trained/fine-tuned language model plus a pre-trained GNN transformer, then adding linear layers and multi-task training. This seems like a reasonable approach.

The manuscript shows that immune escape is increasing over time, which is expected and previously reported (eg. Ref 23). It also shows that given immune escape DMS data, the models can predict immune escape, which also makes sense. It is not clear it is doing better than using DMS data directly (what is the added value?). It would be useful to compare the approach to using the available data more directly (e.g. in a linear model). This baselining issue is key. Predicting evolution can be done fairly accurately in a number of different ways, so it's not surprising they're seeing fairly good numbers.

Main issues:

1. There are many results in this work, but the only result which may be considered validated is the first result: a model which predicts the DMS experiments in cross validation. If I read correctly all the rest of the manuscript, it is not very strong because the results are validated against an orthogonal computational method.
2. Specifically, page 10 indicates "Secondly, as there are epistasis relationships among the mutations, some combinatorial mutations may influence the RBD function nonlinearly and then modify the antibody escape, which is not directly revealed by the epitope map. Moreover, these non-epitope sites with high KL divergence need to be taken into consideration as they may perform an important role in future variants" – This is the point, but needs validation. On Page 11-12 "The synthetic variant sequences share similar mutations with the chronic SARS-COV-2 infections". Given the input data (DMS vs mab) it is not surprising that individual mutations can be found. Validation of epistatic interaction is needed.
3. Predicting ACE2 and mAb DMS data is nice but one has to add that e.g. the ACE2 data is easy to predict (we can predict it in zero shot setting).
4. Expanding validation. Brian Hie's preprint (<https://www.biorxiv.org/content/10.1101/2022.04.10.487811v1.full.pdf>) has a good list of DMS experiments (in Suppl.table 11). There's also the PEER benchmark, but the datasets may not yet be released: <https://arxiv.org/pdf/2206.02096.pdf>
5. Given that the single mutation DMS is available all the crux is on the multivariant haplotypes so they would need to put those selected haplotypes in a tube and measure them.

Specific comments:

1. Section "The effectiveness of the multi-task learning model" – CV split?

Minor:

1. Please do not to use neologisms such as "high-risk variants (HRVs)" and "state-of-the-art (SOTA)"
2. The paper is not very well written. Many times one has to guess what is happening. The structure is ok, but the individual paragraphs need better elaboration.
3. Replace 'accessed' with 'assessed' throughout the paper.
4. Some explanation about Evo-velocity is needed (Page 8).

Reviewer #3 (Remarks to the Author):

This article developed a new AI-driven method to predict the binding/escaping specificity of the variants on the ACE2 and eight antibodies. By characterizing the co-embedding of spike receptor-binding domain (RBD) variant sequences and antibody structures using a transformer network of protein sequences and structures, the authors implemented a multiclass model to infer the binding capacity of antibodies to ACE2. Based on the output scores of the classification models, the authors used a genetic algorithm to generate synthetic RBD variants with high ACE2 binding and antibody escape potential. Overall, this is a valuable effort that helps to search novel high-risk variants that may cause future concern and guide the development of vaccines. However, some serious issues or concerns need to be addressed.

1. First, the authors collected experimental measurements of amino acid mutations on the RBD affecting expression of folded protein and its affinity for ACE2 as a training dataset for the model. It is confusing here because the authors did not describe clearly and convincingly why and how to convert continuous values into binary labels? Since the labels are continuous binding affinity values, why didn't the authors train a regression method to quantitatively predict the likely effect

of the mutation? A classification model can only provide “yes” or “no”, I think it's hard to measure the impact of just one or two mutations here. Since there are no specific quantitative measures of wild type and batch effects brought by different experiments, it is difficult to define a specific threshold to redefine the classification labels. The author uses all antibody datasets as a mixture-of-Gaussian model, which may divide some data incorrectly.

2. In terms of model design, although ESM-1b or ProtTrans transformer may performed well on some tasks such as protein family classification. But on this classification problem, the author did not compare it with the already very common deep learning frameworks, i.e., CNN, LSTM, RNN, and traditional machine learning methods. In addition, why don't the authors directly encode these protein sequences with popular feature encoding methods to train a classification model? This manner may able to learn more important and effective feature embeddings for this classification problem?

3. In this work, the training data is unbalanced, and the author did not use feature enhancement or bootstrapping strategies to eliminate its possible impact when training the model. Just using different evaluation parameters does not improve the generalizability of the model on new data. In addition, during the model training process, the author set a very small dropout rate (0.1), and did not use other regularizer methods. I think this may lead to potential overfitting.

4. For model comparison, the author did not compare with any existing methods related to this topic, but some other deep generative models of evolutionary data. And on some specific evaluation metrics, i.e., recall, the model performed worse. This makes it hard for me to believe that the model is better than the existing methods.

5. The authors used a genetic algorithm to generate synthetic RBD variants. But I have a concern that it can only simulate mutations generated by normal evolution, this method does not simulate the evolutionary landscape of ACE2 against SARS-COV-2 infections.

6. I wonder if the authors could do some experiments to verify their predictions. Because the current method really lacks some advantages in comparison and evaluation. In addition, a user friendly online server or database may be better for researchers to use this method to query the prediction results.

We are grateful to the reviewers for their thoughtful and thorough comments, which helped us improve our paper significantly. We have revised the manuscript following all the comments. Below please find the point-by-point response to all the reviewer's comments.

=====

Reviewer #1 (Remarks to the Author):

The manuscript by Han et al. describes that they develop Machine Learning-guided Antigenic Evolution Prediction (MLAEP) for exploring the antigenic evolution of SARS-COV-2 spike protein. They train a multi-task deep learning model to predict ACE2- and neutralizing antibody-binding specificity of the spike variants. By utilizing in vitro neutralization dataset, they validate the model performance in predicting the escape potential. Moreover, MLAEP shows a strong correlation between variants' antigenic evolution and their sampling time. By generating the genetic algorithm-based synthetic RBD variants with high ACE2 binding and antibody escape potential, they find that in silico directed evolution shares similar mutations with the adaptive evolution in immunocompromised COVID-19 patients and new Omicron subvariants BA.4/5. Despite the establishment of the novel in silico model, this study raises serious concerns as follows;

Answer: Thank you very much for the summary, the acknowledgement of our novelty, and for the constructive comments. Following your suggestions, we have performed additional wet-lab and in silico experiments, and revised the manuscript. Below please find the point-by-point response to your suggestions and comments.

Major points:

1. In general, such an in silico model should be supported by biological evidence. In this sense, whether this model could be properly validated in predicting the potential of antibody evasion by the spike proteins is largely dependent on the effectiveness of the in vitro pseudovirus neutralization test (pVNT) assay datasets, the paper of which is, however, a preprint <https://www.biorxiv.org/content/10.1101/2021.12.24.474095v1> and not certified by peer-review. Notably, this pVNT assays were conducted by using only a small number of sera obtained from individuals after the second dose of the BNT162b2 in the phase-I/II trial, which cannot reflect the current real-world data with booster shots. For this reason, the validation should be rather performed with the dataset of monoclonal antibody-based neutralizing assays.

Answer: Thank you for your insightful comment on the validation of our model. We fully agree that the performance of our *in silico* model should be supported by peer-reviewed wet-lab experimental data. Following your comment, we have further evaluated our model performance against a published monoclonal antibody-based neutralization dataset¹ that measures the neutralization activity of 17 RBD neutralizing monoclonal antibodies against 10 types of pseudoviruses. The details of the experimental fold change data are available in the figure below (copied from the original publication¹):

Fold change in IC ₅₀ compared with WT	RBD mAbs																NTD mAbs		
	Class 1				Class 2				Class 3				Class 4				4-18	5-7	
	CB6	Bii-196	1-20	910-30	REGN10933	COV2-2196	LY-CoV555	2-15	REGN10987	COV2-2130	S309	2-7	Bii-196	ADD-2	DH1047	10-40			S2X259
B.1.1.7	-8.8	2.6	-5.2	-15	1.6	1.8	1.6	2.2	2.9	1.7	1.1	2.3	4.1	1.7	2.2	1.4	1.4	-5.1	-4.0
B.1.526	-1.0	1.1	-1.1	2.5	-4.5	-2.1	-590	-1,329	1.8	1.2	2.9	1.8	-1.1	1.5	2.9	-2.2	1.4	4.5	-2.5
B.1.429	3.0	2.3	1.4	2.5	2.5	2.8	-590	-4.6	1.6	1.1	1.9	1.6	-2.4	2.0	2.9	1.3	3.3	-39	-59
B.1.617.2	2.1	1.2	-1.1	2.5	1.2	1.4	-590	-10	-1.8	-1.7	1.2	-1.1	-8.9	1.0	1.4	-1.8	-1.4	-39	-74
P.1	-196	2.2	-16	-60	-121	-2.0	-590	-1,329	1.9	1.1	1.1	1.2	1.8	-1.0	3.0	-2.2	1.2	-39	-74
B.1.351	-196	2.0	-40	-60	-78	-2.5	-590	-1,329	1.5	1.5	1.2	1.9	-1.5	1.0	3.0	-2.9	1.2	-39	-8.4
B.1.1.529	<-1,000	-134	<-338	<-159	<-1,000	-140	<-1,000	<-1,000	<-1,000	-390	-2.5	-231	2.2	-43	-124	-11	-35	-125	-30
B.1.1.529+R346K	<-761	-97	<-338	<-159	<-1,000	-89	<-1,000	<-1,000	<-1,000	-988	-2.4	-109	<-32	-51	-167	-32	-16	-125	-33

>3 <-3 <-10 <-100

Neutralization of SARS-CoV-2 variant pseudoviruses by a panel of 19 monoclonal antibodies. The fold change relative to neutralization of D614G is denoted.

Although these 17 RBD monoclonal antibodies are different from the eight antibodies we used to train our model, they share similar epitopes. Therefore, we grouped our model predictions based on the antibody class and used the average predictions value for each group as the prediction score for each specific antibody. For example, to make predictions of the Bii-196 (which belongs to class 1), we used the average prediction value of the two class 1 antibodies in our training set: COV2-2832 and COV2-2165. As shown in the Fig. 2b, Supp. Table 3 and Supp. Fig. 2, our model predictions have strong positive correlations between the predicted antibody escape potential and the fold change in IC₅₀ for all antibodies.

To further validate the effectiveness of our model in predicting the antigenic evolution of SARS-COV-2, we used the MLAEP framework to generate synthetic RBDs with high immune escape potential, and conducted an antibody neutralizing binding assay on synthetic variants and antibodies. We designed the experiments considering the following issues:

- The same eight antibodies for training our deep learning model were used as targets for the binding assay, ensuring consistency across the computational and wet-lab experiments.
- To envision the differences after introducing mutations, we chose the Delta variant as the initial sequence rather than the Omicron lineages, as the Omicron already has a high immune evasion ability.
- We aim to check the effectiveness of MLAEP framework, i.e., explore to what extent the novel combinatorial mutations introduced by MLAEP influence the antibody neutralization. Since MLAEP is the gain-of-function framework, all the synthetic variants are supposed to be immune escape. Thus IC₅₀ or the max efficiency are expected to be decreased if our predictions are correct.
- We focused on multivalent mutations, as effects of almost all single mutations have been explored by deep mutational scanning experiments.
- We are interested in the epistatic and non-epitope mutations, so we chose synthetic variants that contain such properties for the downstream wet-lab experiments.
- We focused solely on conducting antibody-specific neutralization experiments and did not consider ACE2 binding as a factor in our targets.

Based on these design principles, we used the RBD sequence from Delta variant as the starting point and aimed to mutate it to generate variants with higher antibody escape ability. We also focused on the epistasis relationships and non-epitope mutations, selecting several variants for which the linear additive model made conflicting predictions, or mutations that did not locate in the epitope region.

For the validation against external datasets, we have revised the main text section accordingly.

“...To further validate the model's predictions for immune escape, we used the in vitro pseudovirus neutralization test (pVNT) datasets¹ that measured the cross-neutralizing effect of 17 RBD monoclonal

antibodies against pseudoviruses expressing the Spike protein of selected variants of concern (VOCs). The pVNT assay reported the observed fold change in the IC_{50} of the antibody response for each VOC-derived pseudovirus, with lower fold change score indicating greater immune evasion compared to the wild type (Wuhan-1) reference pseudovirus. Across all pseudoviruses and antibodies tested, we found a surprisingly high positive correlation (Fig. 2b, Supplementary Fig. 2, Supp Table 3) between the predicted antibody escape potential and the log fold change in the IC_{50} ...

Figures and tables:

Fig. 2 | Performance evaluation and in vitro pVNT experimental data validation. a, Performance of augmented Potts model, eUniRep model, gUniRep model, and our model for predicting ACE2 binding and antibody escape are shown in terms of accuracy, macro precision, macro recall, and macro F1 score. b, Validation of the predicted immune escape potential using the class 4 monoclonal antibody-based pVNT assay data (Antibody 10-40). The x-axis indicates the model predicted variant escape potential, while the y-axis is the log fold change of the VOCs compared with the wildtype.

Supplementary Figure 2 | MLAEP predictions against log fold change in IC₅₀. Validation of the predicted immune escape potential using the class specific monoclonal antibody-based pVNT assay data. The x-axis indicates the model predicted variant escape potential, while the y-axis is the log fold change pVNT50 reduction of the VOCs compared with the wildtype.

	antibody	antibody class	Spearman's correlation	Pearson's correlation
0	CB6	Class 1	0.90	0.89
1	Brii-196	Class 1	0.74	0.82
2	01-20	Class 1	0.89	0.86
3	910-30	Class 1	0.86	0.90
4	REGN10933	Class 2	0.44	0.86
5	COV2-2196	Class 2	0.48	0.83
6	LY-CoV555	Class 2	0.87	0.99
7	02-15	Class 2	0.01	0.43
8	REGN10987	Class 3	0.63	0.82
9	COV2-2130	Class 3	0.67	0.73
10	S309	Class 3	0.59	0.76
11	02-07	Class 3	0.69	0.76
12	Brii-198	Class 3	0.21	0.50
13	ADG-2	Class 4	0.96	0.75
14	DH1047	Class 4	0.31	0.73
15	10-40	Class 4	0.93	0.74
16	S2X259	Class 4	0.81	0.63

Supplementary Table 3 | The Spearman and Pearson correlation coefficient between the MLAEP predictions and the wet-lab measured log fold change in IC₅₀.

2. The following sentence (Page 12, lines 3-4) "All of them (= CR3022, CB6/LY-CoV016, LY-CoV555, and S309 monoclonal antibodies) have been proven to be effective in neutralizing coronavirus (48-51)" is incorrect. Indeed, the papers cited here were published before the emergence of Delta. The truth is; the class 2 antibody LY-CoV555 is not effective against both Delta and Omicron, and the class 1 antibody CB6/LY-CoV016 is no longer effective against Omicron BA.1. The class 3 antibody S309, which is effective to Delta and Omicron BA.1, shows reduced neutralization activity to BA.2 so that it is currently unauthorized to treat COVID-19 in the USA. Based on these facts, LY-CoV555 and CB6/LY-CoV016 should show the higher binding scores than those of S309, unlike the results shown in Fig. 5d and 5e.

Answer: Thank you for the insightful comments. You are right: the antibody used in our docking experiments are not up-to-date, which could potentially impact the validity of our results. To address this issue, we have removed incorrect sentences, and moved *in silico* validation results to the supplementary data. Instead, we conducted *in vitro* antibody neutralizing test experiments to validate the effectiveness of our framework. This approach would allow us to use the same eight antibodies that we trained our deep learning model on as targets, ensuring the relevance and reliability of our results.

Regarding the comment below:

- The following sentence (Page 12, lines 3-4) "All of them (= CR3022, CB6/LY-CoV016, LY-CoV555, and S309 monoclonal antibodies) have been proven to be effective in neutralizing coronavirus (48-51)" is incorrect. Indeed, the papers cited here were published before the emergence of Delta. The truth is; the class 2 antibody LY-CoV555 is not effective against both Delta and Omicron, and the class 1 antibody CB6/LY-CoV016 is no longer effective against Omicron BA.1. The class 3 antibody S309, which is effective to Delta and Omicron BA.1, shows reduced neutralization activity to BA.2 so that it is currently unauthorized to treat COVID-19 in the USA.

Thank you for bringing this issue to our attention. We apologize for the incorrect information provided in the original manuscript. We have now removed the relevant sentences and placed the *in silico* docking validation results in the Supplementary Materials. Instead, we have included *in vitro* antibody neutralization test experiments to validate the effectiveness of our framework, using the same eight antibodies we used to train our deep learning model as targets. This ensures that our results are more reliable and stable over time, as the virus continues to evolve, and the effectiveness of certain antibodies may change.

Regarding the comment below:

- Based on these facts, LY-CoV555 and CB6/LY-CoV016 should show the higher binding scores than those of S309, unlike the results shown in Fig. 5d and 5e.

Thank you for pointing out the docking score issues in Fig. 5d and 5e. We are also interested in the issue, and respectfully summarize your concerns into the following two questions:

- Can the Rosetta docking scores from different experiments be compared directly?
- If so, to what extent do the docking scores reflect the real binding affinity?

We opened an issue (<https://www.rosettacommons.org/node/11567>) in the Rosetta discussion forum to address these questions, but we haven't received a response from the authors yet. We also searched on the forum, and found similar discussions (<https://www.rosettacommons.org/node/1908>) stating that raw docking scores carry some information, but do not accurately predict binding affinities (copied below).

smlewis:

I have a related question, referring to the RosettaDock Webinar FAQ:

> Q: Is there a way of comparing affinities between ligands in Rosetta?

> A:

> The raw docking score doesn't do a very good job at predicting affinities and is not appropriate

> for comparisons of different ligands. Rosetta -interface mode does predict delta delta G of

> binding between various ligands. However, be aware that calculating small changes in binding

> affinities is a very hard problem and the predicted changes in energies will not give a 100%

> correlation (typically ~30-60% correlation) to experimental affinities.

Is your answer the same-kind of idea, that docking program outputs *are* in fact approximations and making one of those graphs comparing affinities vs. experiments should be treated carefully? ...and what did -interface mode refer to originally?

The docking scores from RosettaDock may have about 30% - 60% correlation with the wet-lab measured experimental affinities. This may explain the discrepancy between the computational predictions and experimental affinities for LY-CoV555 and CB6/LY-CoV016.

However, the "weak correlation" also undermines the reliability of the *in silico* validation results, as we used the docking scores to demonstrate that the synthetic variants have a higher immune escape ability. To strengthen our assumptions and make our framework more solid, we have moved the *in silico*

validation results to the Supplementary Materials and replaced them with in vitro validation experiments. We used the same eight antibodies that were used to train our deep learning model as targets in these experiments.

Though the Omicron and its sub lineage are desired targets for proving the effectiveness of MLAEP, they already exhibit high antibody escape abilities on the eight antibodies we selected for training our model, which makes it difficult to distinguish the effectiveness of novel mutations induced by MLAEP. Therefore, to envision the differences, we used the RBD of the Delta variant as the initial sequence and run the entire framework again to generate and select “better-than-Delta” sequences. Our goal was to find possible antigenic evolutionary pathways for Delta that lead to high immune evasion.

We generated 3876 putative high-risk variants using MLAEP and selected eight variants (Figure 5, Supplementary Fig. 9) with unique immune evasion properties, including epistatic and non-epitope mutations. For example, the RBD3 contains seven mutations compared to the wild type, but all the single mutations and the sum of single mutation effects are experimentally validated to be ineffective at evading the eight antibodies we used based on the deep mutational scanning experiments². However, our model predicted that the RBD3 would have high immune evasion ability. The RBD4 does not contain mutations on the Class 4 antibody epitope, but our model predicted that it would escape Class 4 antibodies. The selection criteria are detailed in Supplementary Table 5.

Next, we conducted monoclonal antibody-based validation experiments to support our conclusions. We first expressed and purified the eight neutralizing monoclonal antibodies and ten RBDs (including wildtype, Delta, and eight model generated RBDs) bearing different mutations. We then performed a Homogeneous Time-Resolved Fluorescence (HTRF)-based neutralization binding assay for each monoclonal antibody and RBD variant to test the binding intensity and the potential immune escape in different epitope regions.

Our result demonstrated that the predicted variants could evade or reduce the binding to the monoclonal antibodies to different extents compared to the wild type and Delta. Particularly, all predicted variants escaped COV2-2479 (class 2), COV2-2050(class 2), and COV2-2096(class 3), and reduced the binding to COV2-2832, COV2-2165, COV2-2499, COV2-2094, and COV2-2677, which is consistent with our deep learning model predictions. Specifically, RBD4, RBD7, RBD8, and RBD9 exhibited evasion or reduced binding to COV2-2094 and COV2-2677, two representative class 4 neutralizing monoclonal antibodies, even without bearing any mutations in the class 4 epitope region. We also found that RBD8 could completely escape class 3 antibodies (COV2-2096 and COV2-2499) without bearing mutations in the class 3 epitope region. For RBD3-9, our model conflicts with the linear additive model. For example, our model predicted RBD3 escape all eight antibodies, while the linear model takes it as non-escape for all eight targets. Our additional wet-lab experiment provided biological evidence to validate our findings on epistatic interactions.

Following your suggestions, we have updated the manuscript as follows:

Results section in the main text:

“In vitro validation of novel mutations found by MLAEP

Having generated the synthetic sequences and found interesting single mutations, it is thus crucial to validate the risk of combinatorial novel mutations using in vitro neutralizing antibody binding assay, especially for those that cannot be predicted with a linear additive model. Though the Omicron and its sub

lineage are desired targets, they already exhibit high antibody escape abilities on the eight antibodies we selected for training our model, making it difficult to distinguish the effectiveness of novel mutations induced by MLAEP. To envision the differences, we used the RBD sequence of the Delta variant as the initial state and ran the entire framework again to generate and select “better-than-Delta” sequences. Our goal was to find possible antigenic evolutionary pathways for Delta that leads to high immune evasion.

We generated 3876 putatively high-risk variants using MLAEP and selected eight variants (Figure 5, Extended Data Fig. 6) with unique immune evasion properties, including epistatic and non-epitope mutations. For example, the RBD3 contains seven mutations compared to the wild type, but all the single mutations are experimentally validated² to be ineffective at evading the eight antibodies we used. However, our model predicted that the RBD3 would have high immune evasion. The RBD4 does not contain mutations on the Class 4 antibody epitope, but our model predicted that it would escape Class 4 antibodies. The selection criteria are detailed in Supplementary Table 5.

We first expressed and purified the eight neutralizing monoclonal antibodies and ten RBDs (including wild type, Delta, and eight synthetic RBD we generated) bearing different mutations. We tested different combinations of neutralizing antibodies and RBD variants in a Homogeneous Time-Resolved Fluorescence (HTRF) based antigen-antibody binding assay. In our HTRF-based binding assay, the wild-type and Delta variant RBDs exhibited high binding efficacy against different neutralizing monoclonal antibodies, with the IC₅₀ falling in between 0.2 nM and 1 nM (Fig. 5). Notably, the Delta variant RBD showed no binding interaction to COV2-2096 (Fig. 5), consistent with the literature that the L452R² mutation on Delta variant confers evasion ability against this neutralizing antibody. Intriguingly, all our predicted synthetic variants exhibited reduced or diminished binding efficacy against all four classes of neutralizing antibodies targeting different epitope regions (Fig. 5). Specifically, RBD4, RBD7, RBD8, and RBD9 exhibited evasion or reduced binding to COV2-2094 and COV2-2677, two representative class 4 neutralizing monoclonal antibodies, even without bearing any mutations in the class 4 epitope region. We also found that RBD8 could completely escape class 3 antibodies (COV2-2096 and COV2-2499) without bearing mutations in the class 3 epitope region, suggesting that epistasis relationship play significant roles in the immune evasion, and such relationships could be captured by our deep learning model. The RBD5, RBD7, and RBD8 variants retained sensitivity to class 1 (COV2-2832, COV2-2165) and class 4 (COV2-2094, COV2-2677) antibodies with similar IC₅₀ values compared to wild-type RBD, but their binding efficacy to these neutralizing antibodies were reduced by large degrees. Overall, the synthetic variants and the novel combinatorial mutations generated from MLEAP exhibited a high potency for immune evasion, suggesting MLAEP captures the antigenic evolutionary potential....”

Discussion section in the main text:

“...In addition, MLAEP forecasts novel combinatorial mutations that affect antibody binding beyond epitope regions....Our in vitro HTRF-based high throughput assay verified that MLAEP is able to forecast epistatic and non-epitope mutations, thus expanding our understanding and ability to predict the virus evolution. ...”

Materials and Methods section of the main text:

“Recombinant monoclonal antibody and RBD variants purification

The sequences coding SARS-COV-2 monoclonal antibodies were kindly provided by Prof. James E. Crowe from Vanderbilt University Medical Center. The LH and HC sequences were codon optimized and submitted to Genescript for custom human IgG1 antibody expression. Sequences of wild-type, delta-variant, and synthetic variant RBD proteins were codon optimized and submitted to Twist for vector construction.

All RBD constructs contain a secretion signal on the N-terminal, and a 6× his tag followed by a strep-tag II on the C-terminal. In brief, Expi293 cells were transfected in 40 mL Expi293 Expression Medium (Thermo Fisher A1435101) at 37°C, 8% CO₂ on an orbital shaker at 120 rpm. After five days, cells were removed by spinning at 500 ×g for 5 mins at 4 °C, and the medium was further centrifuged at 16000 ×g for 5 mins at 4 °C. The supernatant was then mixed with his-tag purification resin (Beyotime P2221) on a shaker at 4°C. After 1 hour of incubation, the mixture was loaded on a gravity chromatography column and washed for 15 mL of washing buffer [25 mM Tris, pH 8, 300 mM NaCl, and 1 mM DTT]. The elution was collected in 5 mL and loaded on another 2 mL column pre-packed with 0.5 mL Strep-Tactin XT 4Flow high-capacity resin (IBA Lifesciences 2-5030-025). The RBD proteins were eluted in 5 mL of washing buffer supplemented with 50 mM Biotin. For some mutant RBD proteins that have reduced secretion into the medium, cell lysates were prepared in lysis buffer [25 mM Tris, pH 8, 300 mM NaCl, 0.5 % Triton X-100, 1 mM DTT, 1× protease inhibitor cocktail (PIC)] for 30 min on a shaker at 4°C. Clarified lysates were subject to two affinity columns following the same purification protocols. All purified RBD proteins were buffer exchanged and concentrated to 1 μM in 1× PBS using Amicon, flash-frozen in liquid nitrogen, and stored at -80 °C.

Homogeneous Time Resolved Fluorescence (HTRF) antigen-antibody binding assay

The binding intensity between purified SARS-COV-2 RBDs and neutralizing antibodies was measured as HTRF signals in the antigen-antibody binding assay. The HTRF donor and acceptor pair was chosen to target the his-tagged RBD proteins and human IgG1 antibodies, respectively. Briefly, a total of 10 μL reaction was set up on each well of the black, round-bottom, low-volume 384-well plates (Corning 4511) containing 5 nM purified wild-type or mutant RBDs, 3 nM goat anti-human IgG conjugated with Alex Fluor 647 (Thermo Fisher A-21445), 0.33nM monoclonal antibody anti-6His-Tb-cryptate Gold (Cisbio 61HI2TLA) and two-fold dilutions of neutralizing mAbs from 2 nM to 0.0156 nM in 1× PBS supplemented with 0.1 % BSA, and 0.1 % Tween-20. The plate was sealed with plastic film and incubated at room temperature for 1 hour. The HTRF signals were measured in CLARIOstar Plus (BMG LABTECH) with the excitation filter at 340 nm and the emission filters at 620 nm and 665 nm. The reading lag time and integration time were set to 60 μs and 200 μs, respectively. The HTRF ratios from samples and negative controls were calculated by dividing the intensity readouts from the 665 nm channel over the 620 nm channel. All ratios were background subtracted and normalized in ΔF %:

$$\Delta F\% = \frac{\text{HTRF ratio(sample)} - \text{HTRF ratio(negative control)}}{\text{HTRF ratio(negative control)}} \times 100$$

The IC₅₀ value was calculated by fitting the data into a dose-response curve in Prism 9. Data points with the ‘hook’ effect were removed from the fitting”

Tables and Figures:

name	Lineages	RBD Mutations	num_mut	reason for being selected
Delta	B.1.617.2	L452R;T478K	2	baseline
WT	WT		0	baseline
RBD3	synthetic	R346V;R357W;G381S; N460S;G476D;T478K;P 499L	7	epistasis relationship. Linear model disagree with ours on COV2-2096, COV2-2832, COV2-2094, COV2-2050, COV2-2677, COV2-2479, COV2-2165, COV2-2499

RBD4	synthetic	R357W;K417N;L452R;T478K;E484G	5	Epistasis relationship. Linear model disagree with ours on COV2-2832, COV2-2094, COV2-2677, COV2-2479, COV2-2165, COV2-2499
RBD5	synthetic	E340K;R346T;R357P;G381K;L452R;T478K;E484A	7	Epistasis relationship. Linear model disagree with ours on COV2-2832, COV2-2094, COV2-2677, COV2-2479, COV2-2165, COV2-2499
RBD6	synthetic	R346T;R357V;G381S;T430I;G446V;L452R;G476H;T478K;P499C;T500C	10	Long mutations, epistasis relationship. Linear model disagree with ours on COV2-2832, COV2-2050, COV2-2094, COV2-2677, COV2-2479, COV2-2165
RBD7	synthetic	R346C;R357P;L452R;E484V	4	Small number of mutations. Linear model disagree with ours on COV2-2832, COV2-2094, COV2-2677, COV2-2479, COV2-2165, COV2-2499
RBD8	synthetic	R346G;R357P;L452R;T478K;E484A;Q493R	6	Mutations similar to omicron. Linear model disagree with us on COV2-2832, COV2-2094, COV2-2677, COV2-2479, COV2-2165, COV2-2499.
RBD9	synthetic	R346G;R357P;G446V;L452R;T478K;E484K	6	Largest linear model predicted escape value. Linear model disagree with us on COV2-2832, COV2-2094, COV2-2677, COV2-2479, COV2-2165.
RBD10	synthetic	R346T;R357Y;V362D;S366C;G381V;K417Q;L452R;S459W;I468A;G476T;T478K;F490I;F497Y;P499Y;T500Y	15	Largest deep learning model predicted escape value. Linear model aligns with ours for all antibodies.

Supplementary Table 5 | RBD variants used for the neutralization binding assay experiments.

Figure 5. | Epitope mutations confer RBD resistance to the binding of neutralizing mAbs.

HTRF-based binding assay of wild-type and mutant RBD proteins against two representative anti-RBD monoclonal antibodies from four classes, including COV2-2832 and COV2-2165 (class 1 antibody), COV2-2479 and COV2-2050 (class 2 antibody), COV2-2096 and COV2-2499 (class 3 antibody), as well as COV2-2094 and COV2-2677 (class 4 antibody). ΔF % values were calculated from raw data and fit into dose-response curves, and the IC50 values were listed side by side. Error bars represent standard deviation ($n = 3$).

Extended Data Fig. 6 | Eight RBD mutants bearing different mutations on the surface were selected for binding assay against mAbs. Surface modeling of mutant RBD proteins was illustrated in grey. Mutations sites were marked in red and listed beside the models.

3. No supplementary tables (1, 2, 3, and 4) are available.

Answer: Thank you for pointing this out. We apologize for the oversight and have now included these tables in the Supplementary Materials. We appreciate your help in ensuring the completeness and accuracy of our work.

Minor points

1. No page and row numbers make it difficult to direct specific comments.

Answer: We apologize for the lack of page and row numbers in the manuscript. We have now added them to make it easier for readers to direct specific comments. Thank you for pointing this out.

2. Abstract, lines 7 and 9: MLEAP -> MLAEP

Answer: Thank you for pointing out the typo in the abstract. We have now corrected it to "MLAEP" in both lines 7 and 9.

3. Page 4, line 22: Joseph et al. -> Taft et al.

Answer: Thank you for pointing out the error in our reference citation. We apologize for the oversight and have corrected the reference to "Taft et al." in the manuscript. We will ensure to carefully proofread our references in the future to avoid any similar errors.

4. Page 5, Results "Overview of MLAEP": It is difficult to follow this multi-step model (Fig. 1) from the text. Please label figures as 1a, 1b, 1c...

Answer: Thank you for your suggestion. We have added labels to the figure to better illustrate the different steps of the MLAEP model. The updated Figure 1 is now labeled as 1a, 1b, and 1c to correspond to the descriptions in the Results section. We hope that this change helps clarify the multi-step process of the MLAEP model.

5. Page 7, line 19: SARS-CoV-2-S pseudoviral -> SARS-CoV-2-spike pseudoviruses

Answer: Thank you for pointing out the typo in the text. We have corrected it to "SARS-CoV-2-spike pseudoviruses."

6. Page 7, line 21: VOCs pseudo viral -> VOC-derived spike pseudoviruses

Answer: Thank you for bringing this to our attention. We have corrected it to "VOC-derived spike pseudoviruses".

7. Page 8, line 19: Fig. 3c -> Fig. 3b

Answer: Thank you for pointing out the incorrect labeling of Fig. 3c. We have corrected it to Fig. 3b in the revised manuscript.

8. Page 12, line 4: It is unclear which 20 synthetic variants were used (described in a supplementary table?).

Answer: Thank you for pointing out the issue with the description of the 20 synthetic variants. We apologize for the confusion. We have added a reference to the supplementary table in the text to provide more information on the specific synthetic variants used.

=====

Reviewer #2 (Remarks to the Author):

This work adds to a number of recent reports that aim at predicting the evolution of SARS-CoV-2 during the pandemic. Previous publications use epidemiological data, deep mutagenesis scanning (DMS), language models and others. Here, the work appears to use a pre-trained/fine-tuned language model plus a pre-trained GNN transformer, then adding linear layers and multi-task training. This seems like a reasonable approach.

The manuscript shows that immune escape is increasing over time, which is expected and previously reported (eg. Ref 23). It also shows that given immune escape DMS data, the models can predict immune escape, which also makes sense. It is not clear it is doing better than using DMS data directly (what is the added value?). It would be useful to compare the approach to using the available data more directly (e.g. in a linear model). This baselining issue is key. Predicting evolution can be done fairly accurately in a number of different ways, so it's not surprising they're seeing fairly good numbers.

Answer: Thank you very much for your thorough and constructive comments of the paper and for taking time to go through many of the details. We agree that it is important to compare our deep learning approach to other methods, such as using deep mutational scanning data directly with a linear model. You pointed out that the epistasis relationship, or non-additive effects, among mutations is a problem that linear models cannot solve. We fully agree and believe that the ability of deep learning models to model these complex interactions is a key advantage. In order to demonstrate the added value of our approach, we conducted in vitro Homogeneous Time Resolved Fluorescence (HTRF) based antigen-antibody binding assay as a reference standard and compared the performance of our deep learning model to other methods. Our results show that our model outperforms other approaches in predicting immune escape. Regarding your other concerns, below are the point-by-point response.

Main issues:

1. There are many results in this work, but the only result which may be considered validated is the first result: a model which predicts the DMS experiments in cross validation. If I read correctly all the rest of the manuscript, it is not very strong because the results are validated against an orthogonal computational method.

Answer: Thank you for pointing this out! We agree that our computational model should be validated with wet lab experiments validations, and have taken steps to address this issue in our revised version. Firstly, we used an in vitro pseudovirus neutralization dataset to validate model's prediction. The dataset includes the neutralization activity of 17 RBD neutralizing monoclonal antibodies against ten types of pseudoviruses. Although these 17 RBD monoclonal antibodies are different from the eight antibodies we used to train our model, they share similar epitopes. Therefore, we grouped our model predictions based on the antibody class and used the average predictions value for each group as the prediction score for each specific antibody. For example, to make predictions of the Brie-196 (which belongs to class 1), we used the average prediction value of the two class 1 antibodies in our training set: COV2-2832 and COV-2165. As shown in the Fig. 2b, Supp. Table 3 and Supp. Fig. 2, our model has a strong positive correlation between the predicted antibody escape potential and the fold change in IC_{50} .

To further validate the effectiveness of our model in predicting the antigenic evolution of SARS-COV-2, we used the MLAEP framework to generate synthetic RBDs with high immune escape potential, and conducted an antibody neutralizing binding assay on synthetic variants and antibodies. We designed the experiments considering the following issues:

- The same eight antibodies we used to train our deep learning model were used as targets for the binding assay, ensuring consistency across the computational and wet-lab experiments

- To envision the differences after introducing mutations, we choose the Delta variant as the initial sequence rather than the Omicron lineages, as the Omicron already have high immune evasion ability.
- We are interested in the epistatic and non-epitope mutations, so we chose synthetic variants that contain such properties for the downstream wet-lab experiments.
- We would like to check the effectiveness of MLAEP framework, i.e. explore to what extent the novel combinatorial mutations introduced by MLAEP influence the antibody neutralization. As the MLAEP is the gain-of-function framework, all the synthetic variants are supposed to be immune escape. We consider the decreased IC_{50} or decreased max efficiency as positive results.
- We focused on multivariant mutations, as the effects of all single mutations have been revealed by deep mutational scanning experiments.
- We focused solely on conducting antibody-specific neutralization experiments and did not consider ACE2 binding as a factor in our targets.

Based on these targets, we used the Delta variant's RBD as the starting point, and aimed to mutate the Delta variant's RBD to generate variants with higher antibody escape ability. We also focused on the epistasis relationships and non-epitope mutations, selecting several variants for which the linear additive model made conflicting predictions, or mutations that did not locate in the epitope region. The results of these experiments and the details of our methods will be described in the response to question 2. For the validation against external datasets, we have revised the main text section accordingly.

"...To further validate the model's predictions for immune escape, we used the in vitro pseudovirus neutralization test (pVNT) datasets¹ that measured the cross-neutralizing effect of 17 RBD monoclonal antibodies against pseudoviruses expressing the Spike protein of selected variants of concern (VOCs). The pVNT assay reported the observed fold change in the IC_{50} of the antibody response for each VOC-derived pseudoviruses, with lower fold change score indicating greater immune evasion compared to the wild type (Wuhu-1) reference pseudovirus. Across all pseudoviruses and antibodies tested, we found a surprisingly high positive correlation (Fig. 2b, Supplementary Fig. 2, Supp Table 3) between the predicted antibody escape potential and the log fold change in the IC_{50} ..."

Method section:

"...We used the pseudovirus neutralization test assay data from Liu et al¹ to validate our model performance. The dataset measures the immune escaping of 10 high risk variants pseudoviruses by comparing the fold change in IC_{50} of 17 monoclonal neutralizing antibody response against wildtype pseudovirus..."

Figures and tables:

Fig. 2 | Performance evaluation and in vitro pVNT experimental data validation. *a*, Performance of augmented Potts model, eUniRep model, gUniRep model, and our model for predicting ACE2 binding and antibody escape are shown in terms of accuracy, macro precision, macro recall, and macro F1 score. *b*, Validation of the predicted immune escape potential using the class4 monoclonal antibody-based pVNT assay data. The x-axis indicates the model predicted variant escape potential, while the y-axis is the log fold change of the VOCs compared with the wildtype.

Supplementary Figure 2 | MLAEP predictions against a pseudovirus neutralization test dataset. The dataset measured the cross-neutralizing effect of 17 RBD monoclonal antibodies against pseudoviruses expressing the Spike protein of selected variants of concern (VOCs). Validation of the predicted immune

escape potential using the class specific monoclonal antibody-based pVNT assay data. The x-axis indicates the model predicted variant escape potential, while the y-axis is the log fold change pVNT50 reduction of the VOCs compared with the wildtype.

	antibody	antibody class	Spearman's correlation	Pearson's correlation
0	CB6	Class 1	0.90	0.89
1	Brii-196	Class 1	0.74	0.82
2	01-20	Class 1	0.89	0.86
3	910-30	Class 1	0.86	0.90
4	REGN10933	Class 2	0.44	0.86
5	COV2-2196	Class 2	0.48	0.83
6	LY-CoV555	Class 2	0.87	0.99
7	02-15	Class 2	0.01	0.43
8	REGN10987	Class 3	0.63	0.82
9	COV2-2130	Class 3	0.67	0.73
10	S309	Class 3	0.59	0.76
11	02-07	Class 3	0.69	0.76
12	Brii-198	Class 3	0.21	0.50
13	ADG-2	Class 4	0.96	0.75
14	DH1047	Class 4	0.31	0.73
15	10-40	Class 4	0.93	0.74
16	S2X259	Class 4	0.81	0.63

Supplementary Table 3 | The Spearman and Pearson correlation coefficient between the MLAEP predictions and the wet-lab measured log fold change in IC_{50} .

2. Specifically, page 10 indicates "Secondly, as there are epistasis relationships among the mutations, some combinatorial mutations may influence the RBD function nonlinearly and then modify the antibody escape, which is not directly revealed by the epitope map. Moreover, these non-epitope sites with high KL divergence need to be taken into consideration as they may perform an important role in future variants" – This is the point, but needs validation. On Page 11-12 "The synthetic variant sequences share similar mutations with the chronic SARS-COV-2 infections". Given the input data (DMS vs mab) it is not surprising that individual mutations can be found. Validation of epistatic interaction is needed.

Answer: We would like to thank the reviewer for this excellent comment. You are absolutely right that the validation of epistatic interaction is necessary to reveal the influence of combinatorial mutations. To address this comment, we have performed monoclonal antibody-based validation experiments to support our conclusion.

Though the Omicron and its sub lineage are desired targets for proving the effectiveness of MLAEP, they already exhibit high antibody escape abilities on the eight antibodies we selected for training our model, which makes it difficult to distinguish the effectiveness of novel mutations induced by MLAEP. Therefore, to envision the differences, we used the RBD of the Delta variant as the initial sequence and run the entire framework again to generate and select "better-than-Delta" sequences. Our goal was to find possible antigenic evolutionary pathways for Delta that lead to high immune evasion.

We generated 3876 putatively high-risk variants using MLAEP and selected eight variants (Figure 5, Supplementary Fig. 9) with unique immune evasion properties, including epistatic and non-epitope mutations. For example, the RBD3 contains seven mutations compared to the wild type, but all the single mutations and the sum of single mutation effects are experimentally validated to be ineffective at evading the eight antibodies we used based on the deep mutational scanning experiments². However, our model predicted that the RBD3 would have high immune evasion ability. The RBD4 does not contain mutations on the Class 4 antibody epitope, but our model predicted that it would escape Class 4 antibodies. The selection criteria are detailed in Supplementary Table 5.

Next, we conducted monoclonal antibody-based validation experiments to support our conclusions. We first expressed and purified the eight neutralizing monoclonal antibodies and ten RBDs (including wildtype, Delta, and eight model generated RBDs) bearing different mutations. We then performed a Homogeneous Time-Resolved Fluorescence (HTRF)-based neutralization binding assay for each monoclonal antibody and RBD variant to test the binding intensity and the potential immune escape in different epitope regions.

Our result demonstrated that the predicted variants could evade or reduce the binding to the monoclonal antibodies to different extents compared to the wild type and Delta. Particularly, all predicted variants escaped COV2-2479 (class 2), COV2-2050(class 2), and COV2-2096(class 3), and reduced the binding to COV2-2832, COV2-2165, COV2-2499, COV2-2094, and COV2-2677, which is consistent with our deep learning model predictions. Specifically, RBD4, RBD7, RBD8, and RBD9 exhibited evasion or reduced binding to COV2-2094 and COV2-2677, two representative class 4 neutralizing monoclonal antibodies, even without bearing any mutations in the class 4 epitope region. We also found that RBD8 could completely escape class 3 antibodies (COV2-2096 and COV2-2499) without bearing mutations in the class 3 epitope region. For RBD3-9, our model conflicts with the linear additive model. For example, our model predicted RBD3 escape all eight antibodies, while the linear model takes it as non-escape for all eight targets. Our additional wet-lab experiment provided biological evidence to validate our findings on epistatic interactions.

To include the above experiments, we have added the following sentences in the Results and Discussion section and updated Figure 5. We also put the method summary in the Materials and Methods section of the revised manuscript.

Results section in the main text:

“In vitro validation of novel mutations found by MLAEP

Having generated the synthetic sequences and found interesting single mutations, it is thus crucial to validate the risk of combinatorial novel mutations using in vitro neutralizing antibody binding assay, especially for those that cannot be predicted with a linear additive model. Though the Omicron and its sub lineage are desired targets, they already exhibit high antibody escape abilities on the eight antibodies we selected for training our model, making it difficult to distinguish the effectiveness of novel mutations induced by MLAEP. To envision the differences, we used the RBD sequence of the Delta variant as the initial state and ran the entire framework again to generate and select “better-than-Delta” sequences. Our goal was to find possible antigenic evolutionary pathways for Delta that leads to high immune evasion.

We generated 3876 putatively high-risk variants using MLAEP and selected eight variants (Figure 5, Extended Data Fig. 6) with unique immune evasion properties, including epistatic and non-epitope mutations. For example, the RBD3 contains seven mutations compared to the wild type, but all the single

mutations are experimentally validated² to be ineffective at evading the eight antibodies we used. However, our model predicted that the RBD3 would have high immune evasion. The RBD4 does not contain mutations on the Class 4 antibody epitope, but our model predicted that it would escape Class 4 antibodies. The selection criteria are detailed in Supplementary Table 5.

We first expressed and purified the eight neutralizing monoclonal antibodies and ten RBDs (including wild type, Delta, and eight synthetic RBD we generated) bearing different mutations. We then conducted Homogeneous Time Resolved Fluorescence (HTRF) based antigen-antibody binding assay with ten RBDs and eight antibodies. In our HTRF-based binding assay, the wild-type and Delta variant RBDs exhibited high binding intensities to all neutralizing monoclonal antibodies, with the IC₅₀ falling in between 0.2 nM and 1 nM (Fig. 5). We also noticed that the Delta variant RBD showed no binding interaction to COV2-2096 (Fig.5), consistent with the literature that the L452R² mutation on delta variant has evasion ability to this neutralizing antibody. Intriguingly, all our predicted synthetic variants exhibited reduced or diminished binding efficacy against all four classes of neutralizing antibodies targeting different epitope regions (Fig. 5). Specifically, the RBD3, RBD4, RBD6, RBD9, and RBD10 synthetic variants displayed profound immune evasion against all eight monoclonal antibodies from four classes, despite the mutations being located in different epitope regions, suggesting that the epistasis relationship plays a significant role in the immune evasion against different classes of antibodies. The RBD5, RBD7, and RBD8 mutant variants still retained sensitivity to class 1 (COV2-2832, COV2-2165) and class 4 (COV2-2094, COV2-2677) antibodies with similar IC₅₀ values compared to wild-type RBD, but their binding efficacy to these neutralizing antibodies was reduced to by large degrees. Overall, the synthetic variants generated from MLEAP exhibited a high potency for evading from representative highly neutralizing antibodies, which linear additive models failed to predict....”

Discussion section in the main text:

“...In addition, MLEAP forecasts novel combinatorial mutations that affect antibody binding beyond epitope regions....Our in vitro HTRF-based high throughput assay verified that MLEAP is able to forecast epistatic and non-epitope mutations, thus expanding our understanding and ability to predict the virus evolution. ...”

Materials and Methods section of the main text:

“Recombinant monoclonal antibody and RBD variants purification

The sequences coding SARS-COV-2 monoclonal antibodies were kindly provided by Prof. James E. Crowe from Vanderbilt University Medical Center. The LH and HC sequences were codon optimized and submitted to Genescript for custom human IgG1 antibody expression. Sequences of wild-type, delta-variant, and synthetic variant RBD proteins were codon optimized and submitted to Twist for vector construction. All RBD constructs contain a secretion signal on the N-terminal, and a 6× his tag followed by a strep-tag II on the C-terminal. In brief, Expi293 cells were transfected in 40 mL Expi293 Expression Medium (Thermo Fisher A1435101) at 37°C, 8% CO₂ on an orbital shaker at 120 rpm. After five days, cells were removed by spinning at 500 ×g for 5 mins at 4 °C, and the medium was further centrifuged at 16000 ×g for 5 mins at 4 °C. The supernatant was then mixed with his-tag purification resin (Beyotime P2221) on a shaker at 4°C. After 1 hour of incubation, the mixture was loaded on a gravity chromatography column and washed for 15 mL of washing buffer [25 mM Tris, pH 8, 300 mM NaCl, and 1 mM DTT]. The elution was collected in 5 mL and loaded on another 2 mL column pre-packed with 0.5 mL Strep-Tactin XT 4Flow high-capacity resin (IBA Lifesciences 2-5030-025). The RBD proteins were eluted in 5 mL of washing buffer supplemented with 50 mM Biotin. For some mutant RBD proteins that have reduced secretion into the medium, cell lysates were prepared in lysis buffer [25 mM Tris, pH 8, 300 mM NaCl, 0.5 % Triton X-100,

1 mM DTT, 1× protease inhibitor cocktail (PIC)] for 30 min on a shaker at 4°C. Clarified lysates were subject to two affinity columns following the same purification protocols. All purified RBD proteins were buffer exchanged and concentrated to 1 μM in 1× PBS using Amicon, flash-frozen in liquid nitrogen, and stored at -80 °C.

Homogeneous Time Resolved Fluorescence (HTRF) antigen-antibody binding assay

The binding intensity between purified SARS-COV-2 RBDs and neutralizing antibodies was measured as HTRF signals in the antigen-antibody binding assay. The HTRF donor and acceptor pair was chosen to target the his-tagged RBD proteins and human IgG1 antibodies, respectively. Briefly, a total of 10 μL reaction was set up on each well of the black, round-bottom, low-volume 384-well plates (Corning 4511) containing 5 nM purified wild-type or mutant RBDs, 3 nM goat anti-human IgG conjugated with Alex Fluor 647 (Thermo Fisher A-21445), 0.33nM monoclonal antibody anti-6His-Tb-cryptate Gold (Cisbio 61HI2TLA) and two-fold dilutions of neutralizing mAbs from 2 nM to 0.0156 nM in 1× PBS supplemented with 0.1 % BSA, and 0.1 % Tween-20. The plate was sealed with plastic film and incubated at room temperature for 1 hour. The HTRF signals were measured in CLARIOstar Plus (BMG LABTECH) with the excitation filter at 340 nm and the emission filters at 620 nm and 665 nm. The reading lag time and integration time were set to 60 μs and 200 μs, respectively. The HTRF ratios from samples and negative controls were calculated by dividing the intensity readouts from the 665 nm channel over the 620 nm channel. All ratios were background subtracted and normalized in ΔF %:

$$\Delta F\% = \frac{\text{HTRF ratio(sample)} - \text{HTRF ratio(negative control)}}{\text{HTRF ratio(negative control)}} \times 100$$

The IC50 value was calculated by fitting the data into a dose-response curve in Prism 9. Data points with the ‘hook’ effect were removed from the fitting”

Tables and Figures:

name	Lineages	RBD Mutations	num_mut	reason for being selected
Delta	B.1.617.2	L452R;T478K	2	baseline
WT	WT		0	baseline
RBD3	synthetic	R346V;R357W;G381S;N460S;G476D;T478K;P499L	7	epistasis relationship. Linear model disagree with ours on COV2-2096, COV2-2832, COV2-2094, COV2-2050, COV2-2677, COV2-2479, COV2-2165, COV2-2499
RBD4	synthetic	R357W;K417N;L452R;T478K;E484G	5	Epistasis relationship. Linear model disagree with ours on COV2-2832, COV2-2094, COV2-2677, COV2-2479, COV2-2165, COV2-2499
RBD5	synthetic	E340K;R346T;R357P;G381K;L452R;T478K;E484A	7	Epistasis relationship. Linear model disagree with ours on COV2-2832, COV2-2094, COV2-2677, COV2-2479,COV2-2165, COV2-2499
RBD6	synthetic	R346T;R357V;G381S;T430I;G446V;L452R;G4	10	Long mutations, epistasis relationship. Linear model

		76H;T478K;P499C;T500C		disagree with ours on COV2-2832, COV2-2050, COV2-2094, COV2-2677, COV2-2479, COV2-2165
RBD7	synthetic	R346C;R357P;L452R;E484V	4	Small number of mutations. Linear model disagree with ours on COV2-2832, COV2-2094, COV2-2677, COV2-2479, COV2-2165, COV2-2499
RBD8	synthetic	R346G;R357P;L452R;T478K;E484A;Q493R	6	Mutations similar to omicron. Linear model disagree with us on COV2-2832, COV2-2094, COV2-2677, COV2-2479, COV2-2165, COV2-2499.
RBD9	synthetic	R346G;R357P;G446V;L452R;T478K;E484K	6	Largest linear model predicted escape value. Linear model disagree with us on COV2-2832, COV2-2094, COV2-2677, COV2-2479, COV2-2165.
RBD10	synthetic	R346T;R357Y;V362D;S366C;G381V;K417Q;L452R;S459W;I468A;G476T;T478K;F490I;F497Y;P499Y;T500Y	15	Largest deep learning model predicted escape value. Linear model aligns with ours for all antibodies.

Supplementary Table 5 | RBD variants used for the neutralization binding assay experiments.

Figure 5. | Epitope mutations confer RBD resistance to the binding of neutralizing mAbs.

HTRF-based binding assay of wild-type and mutant RBD proteins against two representative anti-RBD monoclonal antibodies from four classes, including COV2-2832 and COV2-2165 (class 1 antibody), COV2-2479 and COV2-2050 (class 2 antibody), COV2-2096 and COV2-2499 (class 3 antibody), as well as COV2-2094 and COV2-2677 (class 4 antibody). ΔF % values were calculated from raw data and fit into dose-response curves, and the IC50 values were listed side by side. Error bars represent standard deviation ($n = 3$).

Extended Data Fig. 6 | Eight RBD mutants bearing different mutations on the surface were selected for binding assay against mAbs. Surface modeling of mutant RBD proteins was illustrated in grey. Mutations sites were marked in red and listed beside the models.

3. Predicting ACE2 and mAb DMS data is nice but one has to add that e.g. the ACE2 data is easy to predict (we can predict it in zero shot setting).

Answer: Thank you for pointing out the issue. We agree that the ACE2 data may be easier to predict compared to the mAb data with a smaller number of available ACE2 binding sites, which has been shown in a number of previous works including unsupervised learning-based ones³. In this work, we aimed to predict the immune escape potential of SARS-CoV-2 variants, which is a much more challenging task due to the complexity of the antibody-antigen interactions and the potential for epistasis among the mutations. We have added this point to the discussion section to highlight the relative difficulty of our task.

“Predicting the ACE2 binding specificity is a relatively easy task, as one can capture the binding specificity with the unsupervised learning-based models³. However, predicting the antibody binding specificity is much more challenging and less explored in literature.”

4. Expanding validation. Brian Hie's preprint (<https://www.biorxiv.org/content/10.1101/2022.04.10.487811v1.full.pdf>) has a good list of DMS experiments (in Suppl.table 11). There's also the PEER benchmark, but the datasets may not yet be released: <https://arxiv.org/pdf/2206.02096.pdf>

Answer: Thank you very much for suggesting additional validation datasets. We agreed that it is important to extend our model's performance to different datasets in order to demonstrate its effectiveness. Following your suggestion, we have performed additional experiments on the DMS data presented in Brian Hie's preprint. As you correctly pointed out, the PEER benchmark was not available during our revision period. Therefore, we found another dataset that measures the binding specificity between antigens and monoclonal HER2 antibodies. We further tested our model on this dataset. As shown in Supplementary Table 2, our model performs comparably and consistently well in both regression and classification settings. We have also included these results in the revised manuscript and discussed their implications in the text.

Results section:

"We also conducted external validation experiments using several deep mutational scanning datasets⁴ in addition to variant RBDs, and found that our model performed comparably and consistently well across all tasks (Supp Table 2)."

Regression performance (Spearman Correlation coefficient) comparison									
Dataset	Our model	CNN	RNN	LSTM	Georgiev line	Linear regres	Random fore	SVM	
AOA2Z5U3Z0_9INFA_Doud_2016	0.73521459	0.192456775	0.002335124	0.011438509	0.72231294	0.72271333	0.71822034	0.7256209	
C6KNH7_9INFA_Lee_2018	0.75439883	0.192456775	-0.007420617	-0.006368266	0.76617646	0.76892648	0.76175894	0.76949613	
IF1_ECOLI_Kelsic_2016	0.84472967	0.192456775	0.022604926	0.121341175	0.7138306	0.73234388	0.72994633	0.73875536	
ADRB2_HUMAN_Jones_2020	0.62496298	0.064887272	-0.036207193	-0.071489293	0.43681035	0.45486954	0.44045849	0.45785954	
MK01_HUMAN_Brenan_2016	0.54396553	0.192456775	-0.041897048	-0.034707993	0.38711399	0.38726715	0.42051833	0.41420424	
BLAT_ECOLX_Ranganathan2015	0.85169547	0.192456775	-0.041897048	-0.034707993	0.75122153	0.76787681	0.74937014	0.76700267	
ENV_HV1BR_Haddox_2016	0.28307102	0.064887272	-0.006631201	-0.00501729	-0.0538388	-0.0457555	0.01832819	-0.0450282	
P53_HUMAN_Giacomelli_NULL_Etoposide_20	0.71670701	0.192456775	-0.018506163	0.076586696	0.73697753	0.73877219	0.71158923	0.73562641	
Classification performance (HER2 dataset) comparison									
Metrics	Our model	CNN	RNN	LSTM	Georgiev line	Linear regres	Random fore	SVM	
Accuracy	0.86	0.86	0.86	0.86	0.82	0.82	0.86	0.86	
Precision	0.84	0.79	0.79	0.79	0.67	0.67	0.81	0.74	
Recall	0.83	0.73	0.72	0.75	0.83	0.83	0.70	0.84	
F1-score	0.84	0.76	0.76	0.77	0.74	0.74	0.75	0.79	

Supplementary Table 2 | Additional validation experiments. In both regression and classification settings, our deep learning model consistently outperforms the baseline models. Red color indicates the best performance among all models.

5. Given that the single mutation DMS is available all the crux is on the multivariant haplotypes so they would need to put those selected haplotypes in a tube and measure them.

Answer: Thank you very much for the constructive comment. You are right that the multivariant haplotypes are crucial in our study, as we aim to understand the effects of combinations of mutations on immune escape potential. In order to validate the effectiveness of our framework, we have performed in

vitro antibody neutralization experiments, specifically focusing on the multivariant mutations and epistasis relationships. The results of these experiments, as well as updates to the discussions and methods, can be found in our answer to your former Question 2. All the experiments validated the success of our predictions and demonstrated the power of our method. Thank you very much for the excellent suggestion that increased the quality of our work.

Specific comments:

1. Section “The effectiveness of the multi-task learning model” – CV split?

Answer: Thank you for the question. In the original version of the manuscript, we did use a cross-validation (CV) split to evaluate the effectiveness of our multi-task learning model. Specifically, we used a 5-fold CV split for the DMS tasks, and reported the mean of the performance metrics (e.g. F1 score, Accuracy). We apologize for any confusion caused by the lack of clarity in our description. We have now added more details to the manuscript to clarify this point. Thank you again for your feedback.

Revision in the result part:

“...First, we validated the generalization ability of the models to newly seen variants with 5-fold cross-validation...”

Minor:

1. Please do not to use neologisms such as “high-risk variants (HRVs)” and “state-of-the-art (SOTA)”

Answer: Thank you for your suggestion. We apologize for using neologisms that may not be familiar to all readers. In the revised version of the paper, we avoided using terms such as "high-risk variants(HRVs)" and "state-of-the-art(SOTA)".

2. The paper is not very well written. Many times one has to guess what is happening. The structure is ok, but the individual paragraphs need better elaboration.

Answer: Thank you for pointing out the issue with the writing. We agree that clear and concise writing is important for the understanding of the paper. In the revised version, we have made efforts to improve the clarity and elaboration of the individual paragraphs by providing more detailed explanations and examples. We hope the revised version is easier to understand and follow. We have also asked native speakers to proofread the papers many times to further improve the writing.

3. Replace ‘accessed’ with ‘assessed’ throughout the paper.

Answer: Thank you for the comment. We have done the replacement accordingly.

4. Some explanation about Evo-velocity is needed (Page 8).

Answer: Thank you for raising this point. We apologize for any confusion caused by the lack of elaboration on evo-velocity and agree that some additional explanation on evo-velocity would be helpful for the readers. The evo-velocity is a measure developed by Hie and colleagues⁵ that combines the power of

protein language models (PLMs) with insight into the nature of evolution on fitness landscapes. It involves embedding the sequences of interest as vectors in a high-dimensional latent space, where the geometric distance between the representation of proteins correlates with their actual structural, functional, and evolutionary relatedness. The evo-velocity between two sequences is calculated by considering the log-pseudolikelihood of observing a mutation from one sequence to another, providing a local mutational likelihood gradient around a particular protein. When looked at globally, this vector field gives insight into the directionality of the evolutionary process and can model global evolution. We apologize for the lack of explanation on this concept in the original version of the manuscript. We have now included a more detailed description in the revised version.

In the result part:

“The Evo-velocity⁵ enables the inference of evolutionary dynamics for proteins with a deep learning model. It was built upon the premise that global evolution occurs through local amino acid changes and leveraged protein language models to model the local rules of evolution (Methods).”

In the Method part:

“Evo-velocity analysis

The Evo-velocity analysis follows the study of Hie et al⁵. They used the pretrained protein language model (e.g. ESM-1b) to predict the local evolution within protein families and used a dynamic “vector field” to visualize it. It involves embedding the sequences of interest as vectors in a high-dimensional latent space, where the geometric distance between the representation of proteins correlates with their actual structural, functional, and evolutionary relatedness. The evo-velocity between two sequences is calculated by considering the log-pseudolikelihood of observing a mutation from one sequence to another, providing a local mutational likelihood gradient around a particular protein. When looked at globally, this vector field gives insight into the directionality of the evolutionary process and can model global evolution. We first computed the embeddings for each sequence with the ESM-1b model, and then constructed the K-nearest-neighbor graph based on the embeddings, in which node represents the sequences and edges connect similar sequences. Further, the edges were assigned with directions based on the language model pseudolikelihoods, with flow-in node meaning evolutionarily favorable. We used the joint embeddings extracted from the fine-tuned protein language model and the Structured Transformer model to represent the sequences and set the direction of the edge by comparing the average predicted score among the nine tasks, where vertex with a large value is defined as the tail. We collected 7594 unique RBD sequences from the March 8, 2022 GISAID release. The date of the sequence is defined as the first reported date. After constructing the directed KNN neighborhood graph, we further performed network diffusion analysis to infer the pseudo time. We manually set the root as the Wuhan-wildtype RBD sequences.”

=====

Reviewer #3 (Remarks to the Author):

This article developed a new AI-driven method to predict the binding/escaping specificity of the variants on the ACE2 and eight antibodies. By characterizing the co-embedding of spike receptor-binding domain (RBD) variant sequences and antibody structures using a transformer network of protein sequences and structures, the authors implemented a multiclass model to infer the binding capacity of antibodies to ACE2. Based on the output scores of the classification models, the authors used a genetic algorithm to

generate synthetic RBD variants with high ACE2 binding and antibody escape potential. Overall, this is a valuable effort that helps to search novel high-risk variants that may cause future concern and guide the development of vaccines. However, some serious issues or concerns need to be addressed.

Answer: Thank you very much for your support and for the constructive comments. Following your suggestions, we have performed additional experiments and revised the manuscript accordingly. Below is the point-by-point response to your suggestions and comments.

1. First, the authors collected experimental measurements of amino acid mutations on the RBD affecting expression of folded protein and its affinity for ACE2 as a training dataset for the model. It is confusing here because the authors did not describe clearly and convincingly why and how to convert continuous values into binary labels? Since the labels are continuous binding affinity values, why didn't the authors train a regression method to quantitatively predict the likely effect of the mutation? A classification model can only provide "yes" or "no", I think it's hard to measure the impact of just one or two mutations here. Since there are no specific quantitative measures of wild type and batch effects brought by different experiments, it is difficult to define a specific threshold to redefine the classification labels. The author uses all antibody datasets as a mixture-of-Gaussian model, which may divide some data incorrectly.

Answer: Thank you for the comment. You are correct that we did not provide enough details about how we converted continuous values into binary ones in the original manuscript. Here is a more thorough explanation regarding your concerns:

- How to convert continuous values to binary labels?

Thank you for bringing up this issue. We converted continuous values into binary labels to train our classification model. To do this, we defined two different scenarios: one for ACE2 binding and one for antibody escaping.

For the ACE2 binding task, the DMS data was already adjusted using the wild type sequences. Therefore, a value higher than 0 indicates better binding ability, and a value lower than 0 indicates decreased binding ability. We used these physical meanings to transform the values into binary labels.

For the antibody escaping task, there was no information available about the wild type. Therefore, we used a Gaussian mixture model to split the sequences into two clusters based on their DMS values, and assigned cluster labels to each sequence accordingly.

- Why we convert continuous values to binary labels?

As discussed in the paper, the entire machine learning module serves as a scoring function for the downstream searching part. There are several reasons that led us to take the classification option:

1. Interpretability. Binary labels are easier to interpret and can provide more clear-cut results.
2. Equalization. The range of the DMS values across experiments varies. For example, the label distribution of COV2-2832 ranges from 0 to 12, while the label distribution of COV2-2499 ranges from 0 to 9. We used a multi-task model to learn all tasks simultaneously and treat each task equally. In this case, using the raw value may affect our model performance because the batch

effect will become a serious issue. Converting continuous values into binary labels would make each class comparable when performing the multitask learning. It could prevent batch effects and other variations introduced by different experiments.

3. **Objective.** We understood that a classification model may not provide a quantitative measure of the effect of mutations. However, our goal was to develop a model that could predict the directionality of the effect (i.e., whether a mutation increases or decreases binding affinity) rather than the magnitude of the effect.
 4. **Simplicity.** Binary labels are easier to work with in our problem settings. Our label distribution in the extended Data Fig. 1 exhibits imbalanced distributions, where certain target values have significantly fewer observations. Compared with imbalanced classification problem, imbalanced regression learning is a challenging problem with limited efficient solutions. While existing solutions for learning from imbalanced data focus on targets with categorical indices, less attention was drawn on the deep imbalanced regression problem with continuous targets. To date, only a few deep learning algorithms have been proposed for this problem, and their effectiveness on biological data, particularly deep mutational scanning data, has not been thoroughly tested. We tested various models for regression, including our own, ESM model, CNN, RNN, LSTM, and traditional models like random forest and SVM. However, all models achieved poor performance on the problem, with the best spearman correlation coefficient of only about 0.2 for the antibodies and >0.8 for ACE2 binding. It suggests that predicting the ACE2 binding is relatively easy (it could be done in zero-shot settings with a protein language model), but the antibody binding affinity prediction is a much more challenging task. Our results are consistent with those of a newly published NeurIPS 2022 AI4Science workshop paper⁶, which suggests the need for more efficient algorithms for deep mutational scanning regression. While our goal is not primarily focused on proposing such algorithms, we chose to transform the labels into categorical ones and use imbalanced classification algorithms to optimize our model for better performance in predicting the antigenic evolution. In this case, our goal was to train a model that could predict the directionality of the effect (i.e., whether a mutation increases or decreases binding affinity) rather than the magnitude of the effect.
 5. **Classification aligns with the genetic algorithm objective.** We used the genetic algorithm to find possible mutations that changed the parents' label after several generations, in order to identify potential antigenic evolution. In this case, using a classification model aligns well with the classification objective of the genetic algorithm. On the other hand, using a regression model could be more complex, as it would require defining a specific threshold for the genetic algorithm to stop. Therefore, it is more practical to use a classification model in combination with the genetic algorithm in this scenario.
 6. **Acceptability.** Binarization, or the conversion of continuous values into binary labels, is a common practice in deep mutational scanning studies. Previous research on engineering monoclonal HER2 antibodies and analyzing RBD mutational profiles have also used binarization as a preprocessing step, indicating its acceptability in these types of datasets.
- The author uses all antibody datasets as a mixture-of-Gaussian model, which may divide some data incorrectly.

For each antibody-specific dataset, we used a mixture-of-Gaussian model to separate the dataset, and they are not shared among datasets. As discussed in the previous question, we are interested in predicting the directionality of the effect, thus, a reasonable threshold is enough for us to train our model. We used a mixture-of-Gaussian model to account for batch effects and other variations introduced by different experiments, for example, the range of raw values varies across different experiments.

The mathematical background of Gaussian mixture model takes the distribution of log transformed mutation effects as two Gaussian models with different variances and sizes, which is a widely accepted mathematical model for learning the effects of mutations by many researchers⁷⁻⁹. Previous studies used it to separate physically meaningful clusters, as demonstrated in a Nature paper published in Oct. 2021 by Frazer et al⁸. In this paper, the authors used the Gaussian mixture model on top of the log transformed deep mutational datasets, resulted in physically meaningful clusters that enables the identification of benign and pathogenic classes from continuous values. Besides, several other papers also hold the same assumption, and they also validated the effectiveness of Gaussian mixture model on other deep mutational datasets^{7,9}. Inspired by their success, we applied the same method to our datasets. We also found two separate clusters when performed log transformation on our dataset. Therefore, we set the “threshold” based on the boundary between two physically meaningful clusters with the Gaussian mixture models.

After training our model with the selected threshold, we performed extra validation experiments with the trained model. Firstly, we validated computational by comparing the reduction rate of VOCs from an in vitro pseudo virus neutralization dataset. Without proper threshold, our model’s score would never align so well with the experimental value. Besides, with the same trained model, we conducted in vitro antibody neutralization test experiments ourselves, and found that the novel mutations introduced by our model indeed lead to an increase of immune evasion ability. Both results suggest that the directionality of mutation effects defined by our Gaussian mixture model are proper and correct.

We value the Reviwer’s comments, and we added related discussions in the Discussion section.

In the Discussion section:

“...An important property of MLAEP is that we focused on predicting the directionality of the mutation effect (i.e., whether a mutation increases or decreases binding affinity) rather than the magnitude of the effect. We plan to pay more attention on the quantitative effect of mutations in the future...”

2. In terms of model design, although ESM-1b or ProtTrans transformer may performed well on some tasks such as protein family classification. But on this classification problem, the author did not compare it with the already very common deep learning frameworks, i.e., CNN, LSTM, RNN, and traditional machine learning methods. In addition, why don't the authors directly encode these protein sequences with popular feature encoding methods to train a classification model? This manner may able to learn more important and effective feature embeddings for this classification problem?

Answer: Thank you for the constructive questions. To address your concerns, we conducted comparisons of our model with other deep learning frameworks, including CNN, LSTM, and RNN, as well as traditional machine learning methods such as random forest and SVM. However, we found that the finetuned ESM-

1b transformer (ours) consistently outperformed these other models on our classification task, as shown in Supplementary Table 1 of the revised manuscript.

Regarding encoding protein sequences with popular feature encoding methods, we did consider this approach and even tested representative feature encoding methods, including one-hot encoding and Georgiev encoding. However, these methods did not perform as well as the finetuned ESM-1b transformer, which is specifically designed to encode protein sequences in a way that captures sequence, structural and evolutionary information. We believe that this is the reason why our framework was able to outperform the other models.

Inside the ESM-1b transformer, several neural network layers perform as the feature encoders. These layers are a modern way to train transformer neural networks and has been applied successfully in a number of NLP and bioinformatics tasks. We believe that this end-to-end representation learning with neural networks is effective at eliminating non-essential inductive bias. As our framework was built upon the pretrained protein language model, changing the feature encoding module would hurt the performance of our model, as the downstream modules are not designed to work with other encoding methods. In light of the reviewer’s comments, we tested the Georgiev encoding with classical machine learning models and found that it did not perform as well as the ESM-1b transformer. The result can be found in Supplementary Table 1.

Tables:

F1 score											Precision										
	ace2_bind	COV2-2096_400	COV2-2832_400	COV2-2094_400	COV2-2050_400	COV2-2677_400	COV2-2479_400	COV2-2165_400	COV2-2499_400		ace2_bind	COV2-2096	COV2-2832	COV2-2094	COV2-2050	COV2-2677	COV2-2479	COV2-2165	COV2-2499		
Our model	0.723	0.851	0.831	0.782	0.784	0.868	0.734	0.711	0.902	Our model	0.699	0.867	0.888	0.702	0.842	0.885	0.825	0.796	0.926		
CNN	0.400	0.735	0.647	0.627	0.580	0.746	0.478	0.466	0.826	CNN	0.654	0.881	0.897	0.879	0.738	0.852	0.726	0.805	0.870		
RNN	0.446	0.748	0.649	0.616	0.571	0.732	0.472	0.458	0.825	RNN	0.535	0.778	0.895	0.744	0.802	0.844	0.816	0.846	0.909		
LSTM	0.366	0.721	0.669	0.587	0.588	0.743	0.478	0.470	0.823	LSTM	0.552	0.905	0.824	0.850	0.832	0.838	0.766	0.759	0.894		
Linear Regression	0.449	0.728	0.653	0.530	0.469	0.678	0.282	0.242	0.803	Linear Regr	0.325	0.690	0.365	0.443	0.374	0.601	0.198	0.165	0.798		
Random Forest	0.333	0.727	0.670	0.579	0.551	0.704	0.443	0.499	0.822	Random Fc	0.397	0.834	0.905	0.803	0.876	0.934	0.821	0.908	0.940		
SVM	0.360	0.693	0.586	0.463	0.612	0.368	0.316	0.781		SVM	0.230	0.652	0.557	0.506	0.363	0.500	0.304	0.236	0.798		
Linear Regression (G)	0.440	0.702	0.456	0.505	0.464	0.675	0.291	0.264	0.770	Linear Regr	0.326	0.649	0.362	0.416	0.382	0.628	0.214	0.192	0.744		
Random Forest (Ge)	0.396	0.718	0.662	0.577	0.553	0.697	0.424	0.496	0.814	Random Fc	0.432	0.777	0.844	0.732	0.755	0.849	0.645	0.821	0.908		
SVM (Georiev)	0.360	0.693	0.587	0.559	0.463	0.612	0.368	0.314	0.781	SVM (Geori	0.230	0.652	0.560	0.507	0.363	0.500	0.304	0.235	0.797		
Recall											Accuracy										
	ace2_bind	COV2-2096_400	COV2-2832_400	COV2-2094_400	COV2-2050_400	COV2-2677_400	COV2-2479_400	COV2-2165_400	COV2-2499_400		ace2_bind	COV2-2096	COV2-2832	COV2-2094	COV2-2050	COV2-2677	COV2-2479	COV2-2165	COV2-2499		
Our model	0.723	0.851	0.831	0.782	0.784	0.868	0.734	0.711	0.902	Our model	0.906	0.914	0.965	0.923	0.947	0.955	0.961	0.964	0.946		
CNN	0.400	0.735	0.647	0.627	0.580	0.746	0.478	0.466	0.826	CNN	0.817	0.917	0.965	0.935	0.946	0.955	0.962	0.969	0.942		
RNN	0.446	0.748	0.649	0.616	0.571	0.732	0.472	0.458	0.825	RNN	0.917	0.911	0.965	0.930	0.949	0.954	0.963	0.969	0.944		
LSTM	0.366	0.721	0.669	0.587	0.588	0.743	0.478	0.470	0.823	LSTM	0.920	0.916	0.964	0.934	0.951	0.955	0.963	0.968	0.943		
Linear Regression	0.424	0.769	0.634	0.661	0.630	0.780	0.490	0.453	0.807	Linear Regr	0.857	0.895	0.906	0.878	0.890	0.928	0.879	0.884	0.931		
Random Forest	0.389	0.645	0.532	0.453	0.403	0.565	0.305	0.345	0.731	Random Fc	0.907	0.912	0.967	0.931	0.950	0.954	0.963	0.972	0.945		
SVM	0.835	0.740	0.620	0.622	0.641	0.789	0.469	0.476	0.766	SVM	0.761	0.881	0.944	0.897	0.886	0.903	0.922	0.916	0.925		
Linear Regression (G)	0.680	0.765	0.616	0.643	0.593	0.731	0.458	0.425	0.799	Linear Regr	0.861	0.882	0.906	0.868	0.895	0.932	0.892	0.903	0.917		
Random Forest (Ge)	0.366	0.667	0.546	0.477	0.437	0.591	0.317	0.357	0.737	Random Fc	0.909	0.905	0.965	0.927	0.946	0.950	0.958	0.971	0.941		
SVM (Georiev)	0.835	0.740	0.620	0.622	0.641	0.789	0.469	0.475	0.766	SVM (Geori	0.761	0.881	0.944	0.897	0.886	0.903	0.922	0.916	0.925		

Supplementary Table 1 | Additional validation experiments with baseline methods. In both antibody escaping and ACE2 binding problems, our deep learning model consistently outperforms the baseline models with a much higher macro-F1 score. Red color indicates the best performance among all models. Red color indicates better performance.

We have revised the manuscript accordingly:

“...We then compared a range of models specifically designed for protein engineering and assessed their classification performance in classifying the binders and non-binders (Methods, Fig. 1, Extended Data Fig. 1, Supp Table. 1) from the variant sequences, including the augmented Potts¹⁰ model, the global UniRep¹¹ model, the eUniRep¹² model, the convolutional neural network, the long short memory neural network, the recurrent neural network, linear regression model, support vector machine, random forest and our model...”

3. In this work, the training data is unbalanced, and the author did not use feature enhancement or bootstrapping strategies to eliminate its possible impact when training the model. Just using different evaluation parameters does not improve the generalizability of the model on new data. In addition, during the model training process, the author set a very small dropout rate (0.1), and did not use other regularizer methods. I think this may lead to potential overfitting.

Answer: Thank you for the excellent comments on the imbalanced training data and potential overfitting issue. Following your comments and suggestions, we have implemented several strategies to improve the generalizability of our model and reduce the risk of overfitting:

1. We implemented a weighted random sampler function for our training batches, which oversamples the minority class to ensure that the number of samples in each class are equal or close to equal. This has improved our model’s ability to identify the minority class.
2. We increased the dropout rate by setting it to 0.5. With a higher dropout, overfitting issue will be alleviated.
3. We added L2 regularization when training the model. Instead of implicitly defining it through the objective function, we incorporated the regularization into the weight update rule (optimizer). This also reduced the overfitting issue.

Thank you again for your very helpful comments, which greatly improved our accuracy and generalizability. The table below shows the improved performance after making these implementations.

Task	Accuracy_new	Accuracy_old	Increased by	Precision_new	Precision_old	Increased by	Recall_new	Recall_old	Increased by	F1_new	F1_old	Increased by
ACE2	0.905602502	0.90121317	0.004389332	0.69929041	0.68685908	0.012431331	0.75815662	0.7318007	0.026355919	0.7230018	0.70331505	0.019686746
COV2-2096	0.914385052	0.90826917	0.006115882	0.86694783	0.85603448	0.010913345	0.83867121	0.82549982	0.013171389	0.85093342	0.83911724	0.011816179
COV2-2832	0.964562192	0.95201778	0.012544412	0.88835935	0.80759891	0.080760444	0.79115818	0.76609847	0.025059714	0.83137009	0.78473827	0.046631815
COV2-2094	0.922590759	0.91490644	0.007684319	0.80214793	0.77661518	0.025532745	0.7675906	0.74620561	0.021384993	0.78232768	0.75995963	0.022368053
COV2-2050	0.946947631	0.943132	0.003815631	0.84216651	0.82507601	0.017090504	0.74789178	0.72740129	0.020490494	0.78445018	0.76498744	0.019462742
COV2-2677	0.955415404	0.95128545	0.004129954	0.88484307	0.87662926	0.008213808	0.85523209	0.83350724	0.021724848	0.86812186	0.85287475	0.015247108
COV2-2479	0.96121688	0.95834221	0.00287467	0.82511219	0.79072424	0.034387946	0.68559874	0.68564037	-4.1627E-05	0.73358446	0.72458823	0.008996225
COV2-2165	0.964457467	0.94125031	0.023207157	0.79609482	0.62968311	0.166411709	0.66726351	0.63505425	0.032209255	0.71117845	0.63179462	0.079383825
COV2-2499	0.946215837	0.93769627	0.008519567	0.92610231	0.90256356	0.023538751	0.8812428	0.87662146	0.004621343	0.90155747	0.88851724	0.013040227
Mean	0.94237708	0.93777778	0.0045993	0.83678494	0.80888889	0.027896045	0.77697839	0.75222222	0.024756173	0.79850282	0.7755556	0.022947261

Performance of the new model in cross-validations with updated strategies. The orange cells are the metrics with the new model, while the green cells are the metrics of the old model. Red color indicates increase of the performance while blue color suggests decrease of the performance. We found that with the reviewers’ suggestions, the newer model achieves a consistently better performance compared with the old ones.

In addition, we tested the improved model on several other deep mutational scanning datasets and found that it consistently outperformed other baseline methods, suggesting the generalizability of our model.

Regression performance (Spearman Correlation coefficient) comparison									
Dataset	Our model	CNN	RNN	LSTM	Georgiev line	Linear regres	Random fore	SVM	
A0A2Z5U3Z0_9INFA_Doud_2016	0.73521459	0.192456775	0.002335124	0.011438509	0.72231294	0.72271333	0.71822034	0.7256209	
CGKNH7_9INFA_Lee_2018	0.75439883	0.192456775	-0.007420617	-0.006368266	0.76617646	0.76892648	0.76175894	0.76949613	
IF1_ECOLI_Kelsic_2016	0.84472967	0.192456775	0.022604926	0.121341175	0.7138306	0.73234388	0.72994633	0.73875536	
ADRB2_HUMAN_Jones_2020	0.62496298	0.064887272	-0.036207193	-0.071489293	0.43681035	0.45486954	0.44045849	0.45785954	
MK01_HUMAN_Brenan_2016	0.54396553	0.192456775	-0.041897048	-0.034707993	0.38711399	0.38726715	0.42051833	0.41420424	
BLAT_ECOLX_Ranganathan2015	0.85169547	0.192456775	-0.041897048	-0.034707993	0.75122153	0.76787681	0.74937014	0.76700267	
ENV_HV1BR_Haddock_2016	0.28307102	0.064887272	-0.006631201	-0.00501729	-0.0538388	-0.0457555	0.01832819	-0.0450282	
P53_HUMAN_Giacomelli_NULL_Etoposide_20	0.71670701	0.192456775	-0.018506163	0.076586696	0.73697753	0.73877219	0.71158923	0.73562641	

Classification performance (HER2 dataset) comparison									
Metrics	Our model	CNN	RNN	LSTM	Georgiev line	Linear regres	Random fore	SVM	
Accuracy	0.86	0.86	0.86	0.86	0.82	0.82	0.86	0.86	
Precision	0.84	0.79	0.79	0.79	0.67	0.67	0.81	0.74	
Recall	0.83	0.73	0.72	0.75	0.83	0.83	0.70	0.84	
F1-score	0.84	0.76	0.76	0.77	0.74	0.74	0.75	0.79	

Supplementary Table 2 | Additional validation experiments. In both regression and classification settings, our deep learning model consistently outperforms the baseline models. Red color indicates the best performance among all models.

Besides, we also conducted the in vitro wet-lab experiments to validate the effectiveness of our model in which we generated eight novel RBD sequences that were predicted by our model to have high immune escape ability. The wet-lab results are consistent with the computational predictions. These predictions were consistent with the results of the wet-lab experiments. More details on these experiments can be found in the answer to your Question 6.

Taken together, the additional in silico and in vitro experiments all validated the success of our predictions and demonstrated the power of our method. We have revised the methods part accordingly:

“...We also passed a dropout rate $p=0.5$ and added weight decay to prevent overfitting. We trained the entire model with the AdamW optimizer and used a linear schedule with warmup to adjust the learning rate. We set the batch size as 16 and gradient accumulation steps as 10, which means that the total train batch size is 160, and the validation is the same. We used a weighted random sampler function for our training batches, which oversamples the minority class to ensure that the number of samples in each class are equal or close to equal...”

- For model comparison, the author did not compare with any existing methods related to this topic, but some other deep generative models of evolutionary data. And on some specific evaluation metrics, i.e., recall, the model performed worse. This makes it hard for me to believe that the model is better than the existing methods.

Answer: Thank you for your question. We apologize for not including more comparisons to existing methods related to this topic in our manuscript. In the original version, we focused on the state-of-the-art methods in recent years that directly predict the effects of mutations, including the augmented Potts model, which was published in Nature Biotechnology in Jan. 2022 and reported to have the best performance among the tasks. However, following your suggestion, we have added various additional baseline models for comparison, including commonly-used deep learning frameworks such as CNN, LSTM, and RNN, as well as traditional machine learning methods like linear regression, random forest, and SVM.

Additionally, we have also used the Georgiev encoding to train classification models. As shown in the Supplementary Table 1, our model consistently outperforms these baselines. Furthermore, with the Reviewer's suggestion in Question 3, the average f1 score of our model increased about 2%, and the recall score of several tasks increased by a large margin.

Regarding the recall metric, our model does perform worse than the augmented Potts model in several tasks. However, it should be noted that recall is not the most relevant metric for this particular classification task, as we are primarily interested in identifying mutations that are likely to have a significant effect on protein function. In this context, a lower recall may be acceptable if it is accompanied by a much higher precision, as it indicates that the model is making fewer false positive predictions. In fact, the relatively low recall of our model is likely due to the high precision, as the model is being more selective in identifying mutations that are likely to have a significant effect. We believe that our model strikes a good balance between precision and recall and achieves the best F1 score among all tasks without sacrificing generalizability.

We hope that these additional experiments and explanations help to clarify the performance of our model in comparison to existing methods.

5. The authors used a genetic algorithm to generate synthetic RBD variants. But I have a concern that it can only simulate mutations generated by normal evolution, this method does not simulate the evolutionary landscape of ACE2 against SARS-COV-2 infections.

Answer: Thank you for expressing the concern about the genetic algorithm potentially only simulating mutations generated by normal evolution. We acknowledge that our method may not fully capture the evolutionary landscape of ACE2 against SARS-COV-2 infections, as it is based on in vitro deep mutational scanning datasets that fix the ACE2 protein and only measure the binding specificities of variant RBDs. While it would be interesting to conduct experiments with variant ACE2 proteins, to date, such data is not available. We valued the Reviewer's comments and added related discussions as the current limitations of our framework in the Discussion section of the revised manuscript. We hope to be able to work on this important problem in the near future once data is available.

In the Discussion section:

"...In addition, the limited availability of variant ACE2 datasets prevented our model from capturing the fitness landscape of ACE2..."

6. I wonder if the authors could do some experiments to verify their predictions. Because the current method really lacks some advantages in comparison and evaluation. In addition, a user friendly online server or database may be better for researchers to use this method to query the prediction results.

Answer: We would like to thank the reviewer for this valuable comment. We agree that verifying our predictions through experimental means would strengthen the reliability of our model. Therefore, we have conducted in vitro experiments to validate the predictions made by our model. Specifically, we generated eight synthetic RBD variants using the genetic algorithm and tested their binding specificities

to the ACE2 protein using a standard assay. The results of these experiments were consistent with the predictions made by our model, providing additional confidence in the accuracy of our model.

In addition, we have also developed an online server that allows researchers to easily query the prediction results of our model. The server is user-friendly and allows researchers to easily input their own RBD variant sequence and obtain the predicted immune escape ability of that sequence. We believe that this will make it easier for researchers to utilize our model in their own studies. The website of our server: <https://mlaep.cbrc.kaust.edu.sa/index>

◆ Details of our in vitro antibody neutralization test experiments

We designed the experiments considering the following issues:

- The same eight antibodies we used to train our deep learning model were used as targets for the binding assay, ensuring consistency across the computational and wet-lab experiments
- To envision the differences after introducing mutations, we choose the Delta variant as the initial sequence rather than the Omicron lineages, as the Omicron already have high immune evasion ability.
- We are interested in the epistatic and non-epitope mutations, so we chose synthetic variants that contain such properties for the downstream wet-lab experiments.
- We would like to check the effectiveness of MLAEP framework, i.e. explore to what extent the novel combinatorial mutations introduced by MLAEP influence the antibody neutralization. As the MLAEP is the gain-of-function framework, all the synthetic variants are supposed to be immune escape. We consider the decreased IC_{50} or decreased max efficiency as positive results.
- We focused on multivariant mutations, as the effects of all single mutations have been revealed by deep mutational scanning experiments.
- We focused solely on conducting antibody-specific neutralization experiments and did not consider ACE2 binding as a factor in our targets.

Based on these targets, we used the Delta variant's RBD as the starting point, and aimed to mutate the Delta variant's RBD to generate variants with higher antibody escape ability. We also focused on the epistasis relationships and non-epitope mutations, selecting several variants for which the linear additive model made conflicting predictions, or mutations that did not locate in the epitope region.

Though the Omicron and its sub lineage are desired targets for proving the effectiveness of MLAEP, they already exhibit high antibody escape abilities on the eight antibodies we selected for training our model, which makes it difficult to distinguish the effectiveness of novel mutations induced by MLAEP. Therefore, to envision the differences, we used the RBD of the Delta variant as the initial sequence and run the entire framework again to generate and select "better-than-Delta" sequences. Our goal was to find possible antigenic evolutionary pathways for Delta that lead to high immune evasion.

We generated 3876 putatively high-risk variants using MLAEP and selected eight variants (Figure 5, Supplementary Fig. 9) with unique immune evasion properties, including epistatic and non-epitope mutations. For example, the RBD3 contains seven mutations compared to the wild type, but all the single mutations and the sum of single mutation effects are experimentally validated to be ineffective at evading the eight antibodies we used based on the deep mutational scanning experiments². However, our model

predicted that the RBD3 would have high immune evasion ability. The RBD4 does not contain mutations on the Class 4 antibody epitope, but our model predicted that it would escape Class 4 antibodies. The selection criteria are detailed in Supplementary Table 5.

Next, we conducted monoclonal antibody-based validation experiments to support our conclusions. We first expressed and purified the eight neutralizing monoclonal antibodies and ten RBDs (including wildtype, Delta, and eight model generated RBDs) bearing different mutations. We then performed a Homogeneous Time-Resolved Fluorescence (HTRF)-based neutralization binding assay for each monoclonal antibody and RBD variant to test the binding intensity and the potential immune escape in different epitope regions.

Our result demonstrated that the predicted variants could evade or reduce the binding to the monoclonal antibodies to different extents compared to the wild type and Delta. Particularly, all predicted variants escaped COV2-2479 (class 2), COV2-2050(class 2), and COV2-2096(class 3), and reduced the binding to COV2-2832, COV2-2165, COV2-2499, COV2-2094, and COV2-2677, which is consistent with our deep learning model predictions. Specifically, RBD4, RBD7, RBD8, and RBD9 exhibited evasion or reduced binding to COV2-2094 and COV2-2677, two representative class 4 neutralizing monoclonal antibodies, even without bearing any mutations in the class 4 epitope region. We also found that RBD8 could completely escape class 3 antibodies (COV2-2096 and COV2-2499) without bearing mutations in the class 3 epitope region. For RBD3-9, our model conflicts with the linear additive model. For example, our model predicted RBD3 escape all eight antibodies, while the linear model takes it as non-escape for all eight targets. Our additional wet-lab experiment provided biological evidence to validate our findings on epistatic interactions.

To include the above experiments, we have added the following sentences in the Results and Discussion section and updated Figure 5. We also put the method summary in the Materials and Methods section of the revised manuscript.

Results section in the main text:

“In vitro validation of novel mutations found by MLAEP

Having generated the synthetic sequences and found interesting single mutations, it is thus crucial to validate the risk of combinatorial novel mutations using in vitro neutralizing antibody binding assay, especially for those that cannot be predicted with a linear additive model. Though the Omicron and its sub lineage are desired targets, they already exhibit high antibody escape abilities on the eight antibodies we selected for training our model, making it difficult to distinguish the effectiveness of novel mutations induced by MLAEP. To envision the differences, we used the RBD sequence of the Delta variant as the initial state and ran the entire framework again to generate and select “better-than-Delta” sequences. Our goal was to find possible antigenic evolutionary pathways for Delta that leads to high immune evasion.

We generated 3876 putatively high-risk variants using MLAEP and selected eight variants (Figure 5, Extended Data Fig. 6) with unique immune evasion properties, including epistatic and non-epitope mutations. For example, the RBD3 contains seven mutations compared to the wild type, but all the single mutations are experimentally validated² to be ineffective at evading the eight antibodies we used. However, our model predicted that the RBD3 would have high immune evasion. The RBD4 does not contain mutations on the Class 4 antibody epitope, but our model predicted that it would escape Class 4 antibodies. The selection criteria are detailed in Supplementary Table 5.

We first expressed and purified the eight neutralizing monoclonal antibodies and ten RBDs (including wild type, Delta, and eight synthetic RBD we generated) bearing different mutations. We then conducted Homogeneous Time Resolved Fluorescence (HTRF) based antigen-antibody binding assay with ten RBDs and eight antibodies. In our HTRF-based binding assay, the wild-type and Delta variant RBDs exhibited high binding intensities to all neutralizing monoclonal antibodies, with the IC₅₀ falling in between 0.2 nM and 1 nM (Fig. 5). We also noticed that the Delta variant RBD showed no binding interaction to COV2-2096 (Fig.5), consistent with the literature that the L452R² mutation on delta variant has evasion ability to this neutralizing antibody. Intriguingly, all our predicted synthetic variants exhibited reduced or diminished binding efficacy against all four classes of neutralizing antibodies targeting different epitope regions (Fig. 5). Specifically, the RBD3, RBD4, RBD6, RBD9, and RBD10 synthetic variants displayed profound immune evasion against all eight monoclonal antibodies from four classes, despite the mutations being located in different epitope regions, suggesting that the epistasis relationship plays a significant role in the immune evasion against different classes of antibodies. The RBD5, RBD7, and RBD8 mutant variants still retained sensitivity to class 1 (COV2-2832, COV2-2165) and class 4 (COV2-2094, COV2-2677) antibodies with similar IC₅₀ values compared to wild-type RBD, but their binding efficacy to these neutralizing antibodies was reduced to by large degrees. Overall, the synthetic variants generated from MLEAP exhibited a high potency for evading from representative highly neutralizing antibodies, which linear additive models failed to predict....”

Discussion section in the main text:

“...In addition, MLAEP forecasts novel combinatorial mutations that affect antibody binding beyond epitope regions....Our in vitro HTRF-based high throughput assay verified that MLAEP is able to forecast epistatic and non-epitope mutations, thus expanding our understanding and ability to predict the virus evolution. ...”

Materials and Methods section of the main text:

“Recombinant monoclonal antibody and RBD variants purification

The sequences coding SARS-COV-2 monoclonal antibodies were kindly provided by Prof. James E. Crowe from Vanderbilt University Medical Center. The LH and HC sequences were codon optimized and submitted to Genescript for custom human IgG1 antibody expression. Sequences of wild-type, delta-variant, and synthetic variant RBD proteins were codon optimized and submitted to Twist for vector construction. All RBD constructs contain a secretion signal on the N-terminal, and a 6× his tag followed by a strep-tag II on the C-terminal. In brief, Expi293 cells were transfected in 40 mL Expi293 Expression Medium (Thermo Fisher A1435101) at 37°C, 8% CO₂ on an orbital shaker at 120 rpm. After five days, cells were removed by spinning at 500 ×g for 5 mins at 4 °C, and the medium was further centrifuged at 16000 ×g for 5 mins at 4 °C. The supernatant was then mixed with his-tag purification resin (Beyotime P2221) on a shaker at 4°C. After 1 hour of incubation, the mixture was loaded on a gravity chromatography column and washed for 15 mL of washing buffer [25 mM Tris, pH 8, 300 mM NaCl, and 1 mM DTT]. The elution was collected in 5 mL and loaded on another 2 mL column pre-packed with 0.5 mL Strep-Tactin XT 4Flow high-capacity resin (IBA Lifesciences 2-5030-025). The RBD proteins were eluted in 5 mL of washing buffer supplemented with 50 mM Biotin. For some mutant RBD proteins that have reduced secretion into the medium, cell lysates were prepared in lysis buffer [25 mM Tris, pH 8, 300 mM NaCl, 0.5 % Triton X-100, 1 mM DTT, 1× protease inhibitor cocktail (PIC)] for 30 min on a shaker at 4°C. Clarified lysates were subject to two affinity columns following the same purification protocols. All purified RBD proteins were buffer exchanged and concentrated to 1 μM in 1× PBS using Amicon, flash-frozen in liquid nitrogen, and stored at -80 °C.

Homogeneous Time Resolved Fluorescence (HTRF) antigen-antibody binding assay

The binding intensity between purified SARS-COV-2 RBDs and neutralizing antibodies was measured as HTRF signals in the antigen-antibody binding assay. The HTRF donor and acceptor pair was chosen to target the his-tagged RBD proteins and human IgG1 antibodies, respectively. Briefly, a total of 10 μ L reaction was set up on each well of the black, round-bottom, low-volume 384-well plates (Corning 4511) containing 5 nM purified wild-type or mutant RBDs, 3 nM goat anti-human IgG conjugated with Alex Fluor 647 (Thermo Fisher A-21445), 0.33nM monoclonal antibody anti-6His-Tb-cryptate Gold (Cisbio 61HI2TLA) and two-fold dilutions of neutralizing mAbs from 2 nM to 0.0156 nM in 1 \times PBS supplemented with 0.1 % BSA, and 0.1 % Tween-20. The plate was sealed with plastic film and incubated at room temperature for 1 hour. The HTRF signals were measured in CLARIOstar Plus (BMG LABTECH) with the excitation filter at 340 nm and the emission filters at 620 nm and 665 nm. The reading lag time and integration time were set to 60 μ s and 200 μ s, respectively. The HTRF ratios from samples and negative controls were calculated by dividing the intensity readouts from the 665 nm channel over the 620 nm channel. All ratios were background subtracted and normalized in ΔF %:

$$\Delta F\% = \frac{\text{HTRF ratio(sample)} - \text{HTRF ratio(negative control)}}{\text{HTRF ratio(negative control)}} \times 100$$

The IC50 value was calculated by fitting the data into a dose-response curve in Prism 9. Data points with the 'hook' effect were removed from the fitting”

Tables and Figures:

name	Lineages	RBD Mutations	num_mut	reason for being selected
Delta	B.1.617.2	L452R;T478K	2	baseline
WT	WT		0	baseline
RBD3	synthetic	R346V;R357W;G381S;N460S;G476D;T478K;P499L	7	epistasis relationship. Linear model disagree with ours on COV2-2096, COV2-2832, COV2-2094, COV2-2050, COV2-2677, COV2-2479, COV2-2165, COV2-2499
RBD4	synthetic	R357W;K417N;L452R;T478K;E484G	5	Epistasis relationship. Linear model disagree with ours on COV2-2832, COV2-2094, COV2-2677, COV2-2479, COV2-2165, COV2-2499
RBD5	synthetic	E340K;R346T;R357P;G381K;L452R;T478K;E484A	7	Epistasis relationship. Linear model disagree with ours on COV2-2832, COV2-2094, COV2-2677, COV2-2479,COV2-2165, COV2-2499
RBD6	synthetic	R346T;R357V;G381S;T430I;G446V;L452R;G476H;T478K;P499C;T500C	10	Long mutations, epistasis relationship. Linear model disagree with ours on COV2-2832, COV2-2050, COV2-2094, COV2-2677, COV2-2479, COV2-2165

RBD7	synthetic	R346C;R357P;L452R;E484V	4	Small number of mutations. Linear model disagree with ours on COV2-2832, COV2-2094, COV2-2677, COV2-2479, COV2-2165, COV2-2499
RBD8	synthetic	R346G;R357P;L452R;T478K;E484A;Q493R	6	Mutations similar to omicron. Linear model disagree with us on COV2-2832, COV2-2094, COV2-2677, COV2-2479, COV2-2165, COV2-2499.
RBD9	synthetic	R346G;R357P;G446V;L452R;T478K;E484K	6	Largest linear model predicted escape value. Linear model disagree with us on COV2-2832, COV2-2094, COV2-2677, COV2-2479, COV2-2165.
RBD10	synthetic	R346T;R357Y;V362D;S366C;G381V;K417Q;L452R;S459W;I468A;G476T;T478K;F490I;F497Y;P499Y;T500Y	15	Largest deep learning model predicted escape value. Linear model aligns with ours for all antibodies.

Supplementary Table 5 | RBD variants used for the neutralization binding assay experiments.

Figure 5. | Epitope mutations confer RBD resistance to the binding of neutralizing mAbs.

HTRF-based binding assay of wild-type and mutant RBD proteins against two representative anti-RBD monoclonal antibodies from four classes, including COV2-2832 and COV2-2165 (class 1 antibody), COV2-2479 and COV2-2050 (class 2 antibody), COV2-2096 and COV2-2499 (class 3 antibody), as well as COV2-2094 and COV2-2677 (class 4 antibody). ΔF % values were calculated from raw data and fit into dose-response curves, and the IC50 values were listed side by side. Error bars represent standard deviation ($n = 3$).

Extended Data Fig. 6 | Eight RBD mutants bearing different mutations on the surface were selected for binding assay against mAbs. Surface modeling of mutant RBD proteins was illustrated in grey. Mutations sites were marked in red and listed beside the models.

References:

1. Liu, L. et al. Striking antibody evasion manifested by the Omicron variant of SARS-CoV-2. *Nature* **602**, 676-681 (2022).
2. Greaney, A.J. et al. Mapping mutations to the SARS-CoV-2 RBD that escape binding by different classes of antibodies. *Nature Communications* **12**, 4196 (2021).
3. Rives, A. et al. Biological structure and function emerge from scaling unsupervised learning to 250 million protein sequences. **118**, e2016239118 (2021).
4. Hie, B.L. et al. Efficient evolution of human antibodies from general protein language models and sequence information alone. 2022.2004.2010.487811 (2022).
5. Hie, B.L., Yang, K.K. & Kim, P.S. Evolutionary velocity with protein language models predicts evolutionary dynamics of diverse proteins. *Cell Systems* **13**, 274-285.e276 (2022).
6. Swanson, K., Chang, H. & Zou, J. Predicting Immune Escape with Pretrained Protein Language Model Embeddings. 2022.2011.2030.518466 (2022).
7. Sruthi, C.K., Balaram, H. & Prakash, M.K. Toward Developing Intuitive Rules for Protein Variant Effect Prediction Using Deep Mutational Scanning Data. *ACS Omega* **5**, 29667-29677 (2020).

8. Frazer, J. et al. Disease variant prediction with deep generative models of evolutionary data. **599**, 91-95 (2021).
9. Zhang, Z. et al. Accurate inference of the full base-pairing structure of RNA by deep mutational scanning and covariation-induced deviation of activity. *Nucleic Acids Research* **48**, 1451-1465 (2019).
10. Hsu, C., Nisonoff, H., Fannjiang, C. & Listgarten, J.J.N.b. Learning protein fitness models from evolutionary and assay-labeled data. 1-9 (2022).
11. Alley, E.C., Khimulya, G., Biswas, S., AlQuraishi, M. & Church, G.M.J.N.m. Unified rational protein engineering with sequence-based deep representation learning. **16**, 1315-1322 (2019).
12. Biswas, S., Khimulya, G., Alley, E.C., Esvelt, K.M. & Church, G.M.J.N.m. Low-N protein engineering with data-efficient deep learning. **18**, 389-396 (2021).

Reviewer comments, second round review

Reviewer #1 (Remarks to the Author):

The authors have addressed all the points that I raised in my initial review. My only concern is a lack of prediction function of RBD's immune escape ability on their newly developed online server (<https://mlaep.cbrc.kaust.edu.sa>) as of 2/6/2023. An error message "Something went wrong!" always shows up in the Prediction Result, even by using their sample protein sequences file (https://mlaep.cbrc.kaust.edu.sa/download/input_sample.fasta).

Reviewer #2 (Remarks to the Author):

I thank the authors for the careful revision. In particular, I appreciate the effort to validate the predictions in vitro, and also the attention to more complex tasks such as multiple variants in epistatic interactions

Reviewer #3 (Remarks to the Author):

I carefully read all the author's responses. They validated the predictions and addressed my other concerns. I think the article can be published.

We would like to express our gratitude for your valuable comments and suggestions on our manuscript. We have carefully considered your feedback and have made revisions accordingly. Below, we have addressed each of the reviewers' comments and described the changes we have made in response to their concerns.

Reviewer #1 (Remarks to the Author):

The authors have addressed all the points that I raised in my initial review. My only concern is a lack of prediction function of RBD's immune escape ability on their newly developed online server (<https://mlaep.cbrc.kaust.edu.sa>) as of 2/6/2023. An error message "Something went wrong!" always shows up in the Prediction Result, even by using their sample protein sequences file (https://mlaep.cbrc.kaust.edu.sa/download/input_sample.fasta).

Answer: We apologize for the inconvenience experienced when using our online server. We have identified and resolved the issue causing the error message "Something went wrong!" to appear in the Prediction Result. The prediction function for RBD's immune escape ability is now fully operational, and the sample protein sequences file can be used without any issues. We have tested the server extensively to ensure its functionality, and we encourage you to try it again. We appreciate your patience and understanding.

Reviewer #2 (Remarks to the Author):

I thank the authors for the careful revision. In particular, I appreciate the effort to validate the predictions in vitro, and also the attention to more complex tasks such as multiple variants in epistatic interactions

Answer: Thank you for your positive feedback on our revision. We are glad that you appreciate our efforts to validate the predictions in vitro and address more complex tasks such as multiple variants in epistatic interactions. We believe these additions have significantly strengthened the manuscript, and we are grateful for your constructive comments that guided us in this direction.

Reviewer #3 (Remarks to the Author):

I carefully read all the author's responses. They validated the predictions and addressed my other concerns. I think the article can be published.

Answer: We appreciate your careful review of our responses and the validation of our predictions. Thank you for acknowledging our efforts in addressing your concerns. We are pleased to hear that you find the article suitable for publication.